**EMBO** *reports*

# Heterochromatin-dependent transcription links the PRC2 complex to small RNA-mediated DNA elimination

Therese Solberg [1,2,3]✉, Chundi Wang [1,4,5], Ryuma Matsubara [1,6], Zhiwei Wen[5] & Mariusz Nowacki [1]✉

## Abstract

**Facultative heterochromatin is marked by the repressive histone modification H3K27me3 in eukaryotes. Deposited by the PRC2 complex, H3K27me3 is essential for regulating gene expression during development, and chromatin bearing this mark is generally considered transcriptionally inert. The PRC2 complex has also been linked to programmed DNA elimination during development in ciliates such as *Paramecium*. Due to a lack of mechanistic insight, a direct involvement has been questioned as most eliminated DNA segments in *Paramecium* are shorter than the size of a nucleosome. Here, we identify two sets of histone methylation readers essential for PRC2-mediated DNA elimination in *Paramecium*: Firefly1/2 and Mayfly1-4. The chromodomain proteins Firefly1/2 act in tight association with TFIIS4, a transcription elongation factor required for noncoding RNA transcription. These noncoding transcripts act as scaffolds for sequence-specific targeting by PIWI-bound sRNAs, resulting in local nucleosome depletion and DNA elimination. Our findings elucidate the molecular mechanism underlying the role of PRC2 in PIWI-mediated DNA elimination and suggest that its role in IES elimination may be to activate rather than repress transcription.**

Keywords Heterochromatin; H3K27me3; H3K9me3; PRC2; Small RNA
Subject Categories Chromatin, Transcription & Genomics; RNA Biology

## Introduction

Histone H3 trimethylation of lysine 27 (H3K27me3) and lysine 9 (H3K9me3) are well-known as marks of repressive heterochromatin in various species (Millan-Zambrano et al, 2022). Generally associated with chromatin compaction and inaccessibility to transcription factors and polymerases, chromosome regions decorated with such marks are considered transcriptionally inert. Facultative heterochromatin silences protein-coding genes during development and is associated with H3K27me3; whereas constitutive heterochromatin represses repetitive sequences such as transposable elements (TEs) and satellite DNA and is associated with H3K9me3. However, recent findings scrutinize their roles as simple repressive marks, providing evidence that they may also activate certain forms of transcription, or switch between coding and noncoding transcription modes. One such example is the transcription of piRNA precursors from dual-strand clusters in the *Drosophila* ovary. TEs must be strictly controlled to prevent the expression of their mRNAs, but at the same time they need to be transcribed to generate piRNA precursors used for their targeted repression. In this system, H3K9me3 set by the histone methyltransferase Eggless/SetDB1, represses TE sequences while allowing bidirectional transcription of piRNA clusters (Rangan et al, 2011; Sienski et al, 2015; Yu et al, 2015). Transcriptional activity within heterochromatin is achieved through the action of the Heterochromatin Protein 1 (HP1) homolog Rhino (Rhi), which specifically binds to H3K9me3 at piRNA clusters (Klattenhoff et al, 2009; Le Thomas et al, 2014; Mohn et al, 2014; Zhang et al, 2014). Together with Deadlock and Cutoff as part of the RDC complex, Rhi licenses Pol II to generate noncoding transcripts by recruiting the transcription initiation factor IIA subunit 1 paralogue Moonshiner (Andersen et al, 2017). Consequently, Rhi licenses the transcription of heterochromatic loci while maintaining the repressive state of the cluster by switching the mode of transcription in favor of noncoding transcription. Small RNA-mediated gene silencing mechanisms using heterochromatin-dependent transcription machineries is not unique to the piRNA pathway or flies, but have been reported in several species, including co-transcriptional gene silencing (CTGS) in fission yeast (Buhler et al, 2006; Djupedal et al, 2005; Kato et al, 2005; Volpe et al, 2002) and the RNA-directed DNA methylation (RdDM) pathway in plants (Law et al, 2013; Wierzbicki et al, 2008; Wierzbicki et al, 2009; Zhang et al, 2013). Thus, small RNA pathways often employ specialized transcription machineries to balance transcription and repression within heterochromatin.

[1]Institute of Cell Biology, University of Bern, Baltzerstrasse 4, 3012 Bern, Switzerland. [2]Department of Molecular Biology, Keio University School of Medicine, 160-8582 Tokyo, Japan. [3]Human Biology Microbiome Quantum Research Center (WPI-Bio2Q), Keio University, 108-8345 Tokyo, Japan. [4]Institute of Evolution & Marine Biodiversity, Ocean University of China, 266003 Qingdao, China. [5]Laboratory of Marine Protozoan Biodiversity & Evolution, Marine College, Shandong University, 264209 Weihai, China. [6]Isotope Science Center, The University of Tokyo, 113-0032 Tokyo, Japan. ✉E-mail: therese.solberg@keio.jp; mariusz.nowacki@unibe.ch

Ciliates are excellent model organisms to study the interplay between small RNAs (sRNAs) and chromatin dynamics. Although unicellular, ciliates separate germline and somatic functions into two distinct nuclei. These nuclei co-exist in a single cytoplasm and sexual reproduction features complex genome rearrangements to generate a new somatic genome from the germline genome (Gao et al, 2023). In the ciliate *Paramecium tetraurelia* (*Paramecium*), these rearrangements include imprecise DNA elimination of evolutionarily young TE copies and precise elimination of about 45,000 short DNA segments derived from more ancient transposon insertions known as Internal Eliminated Sequences (IESs) (Arnaiz et al, 2012). Recently, we and others reported a development-specific PRC2 complex to be essential for DNA elimination in *Paramecium* (Miro-Pina et al, 2022; Wang et al, 2022). The complex was shown to be required for transcriptional repression and elimination of young TEs from the genome, which is dependent on the deposition of both H3K9me3 and H3K27me3 on TE sequences during development (Frapporti et al, 2019). However, we also found the complex to be required for the precise elimination of over 30,000 IESs, including all sRNA-dependent IESs. Most IESs in the *Paramecium* genome are shorter than the length of DNA wrapped around a nucleosome, the shortest of which are only 26 bp (Arnaiz et al, 2012). It has been proposed that repressive chromatin marks may be set locally on IESs and used as a mark to recognize the IESs, yet there is no experimental evidence in support of this hypothesis and it is difficult to imagine that this would provide enough information for precise elimination of IESs as short as 26 bp (Lhuillier-Akakpo et al, 2014). Moreover, when two or more ultra-short IESs are located in close proximity to each other, it would be impossible to mark them individually without including the DNA regions that separate them. Thus, there remains a tantalizing gap in our understanding of the underlying molecular mechanism. We also recently reported the identification of the development-specific nucleosome remodeler ISWI1, demonstrating that IESs are depleted of nucleosomes in an ISWI1-dependent manner, a pre-requisite for DNA elimination (Singh et al, 2022). Our findings that IESs are nucleosome-poor is also in conflict with the proposed role of the PRC2 complex on IES elimination.

Targeting of sequences for elimination is believed to occur through the pairing of sRNAs with nascent transcripts in the developing new soma. In favor of this hypothesis, a transcription elongation factor, TFIIS4, was shown to be required for the production of noncoding RNAs (ncRNAs) in the developing new soma (Maliszewska-Olejniczak et al, 2015). TFIIS4 depletion mimics the effect of triple-silencing of Dcl2/3/5, the Dicer-like enzymes generating development-specific small RNAs (Swart et al, 2017). Despite the central role of ncRNA transcripts in DNA elimination, little is known about how their transcription is regulated.

In this work, we propose a unified model that explains how Piwi-bound sRNAs target IESs for elimination in *Paramecium*. We performed a silencing screen to identify chromodomain proteins involved in the genome rearrangement process and characterized two subfamilies that are essential for zygotic development and DNA elimination: the putative H3K27me3 readers Firefly1/2 (Fire) and the H3K9me3 readers Mayfly1-4 (May). We show that Fire and May proteins are required for the removal of different subsets of IESs, and that both subsets also depend on the PRC2 complex for their removal. Moreover, we found that the PRC2 complex is linked

to ncRNA transcription through a tight association of Fire1/2 and the transcription elongation factor TFIIS4. Our data provide evidence in support of a model mechanistically analogous to Rhi-dependent piRNA cluster transcription in *Drosophila* ovaries. In addition to uncovering the molecular mechanism underlying the debated role of PRC2 on sRNA-mediated DNA elimination, these findings suggest that one or both of the "repressive histone modifications" deposited by PRC2 may act as a transcriptional activator in *Paramecium*. Our findings highlight the broad evolutionary conservation of biologically relevant heterochromatin-dependent transcription and provide key insight into the interplay between small RNAs and chromatin dynamics.

# Results

## Identification of *P. tetraurelia* chromodomain proteins

The role of histone modifications in genome rearrangements in *Paramecium* is continuously being dissected at the molecular level, mainly through studies of histone-modifying enzymes, yet not a single histone methylation reader has been linked to the process (Frapporti et al, 2019; Ignarski et al, 2014; Lhuillier-Akakpo et al, 2014; Miro-Pina et al, 2022; Wang et al, 2022). A complementary approach to studying histone-modifying enzymes is to study proteins that bind to and define chromatin states. This is of particular importance in *Paramecium*, considering that both H3K9me3 and H3K27me3 are developmentally regulated and set by the same machinery (PRC2) (Frapporti et al, 2019; Miro-Pina et al, 2022; Wang et al, 2022). Since it is not possible to decipher the role of each modification individually by studying the PRC2 complex, we sought to identify the reader proteins of these methylation marks to individually assess their roles. In a related ciliate, *Tetrahymena thermophila* (*Tetrahymena*), methyl-lysine reader proteins known as chromodomain proteins have been studied for over two decades and are tightly linked to genome rearrangements (Noto and Mochizuki, 2017; Wiley et al, 2018). Although there are important differences between these processes in *Tetrahymena* and *Paramecium*, recent findings revealed that PRC complexes are essential for genome rearrangements in both organisms (Gao et al, 2023; Miro-Pina et al, 2022; Wang et al, 2022; Xu et al, 2021). To identify chromodomain proteins involved in genome rearrangements in *Paramecium*, we searched for proteins with similarity to the chromodomains (CDs) of 13 *Tetrahymena* chromodomain proteins reported in a previous study (Appendix Table S1) (Wiley et al, 2018). Following this search, all ohnologues (paralogs from whole-genome duplication events) of these candidates were also included regardless of whether or not they had been identified in the initial search. This resulted in the identification of 34 proteins, all of which contain one CD (Fig. 1; Dataset EV1). Only one protein, May1, also had a chromo shadow domain (CSD), a related but highly diverged domain that does not contain the aromatic residues required for binding to methylated lysines (Aasland and Stewart, 1995). Of these 34 genes, 91.2% are upregulated during sexual development (Fig. 1B; Dataset EV1) (Arnaiz et al, 2017). Among them, the majority fall under the category of "Intermediate peak" (23), followed by "Early peak" (4) and "Late induction" (4). Only three are not developmentally regulated. The large number of developmentally regulated

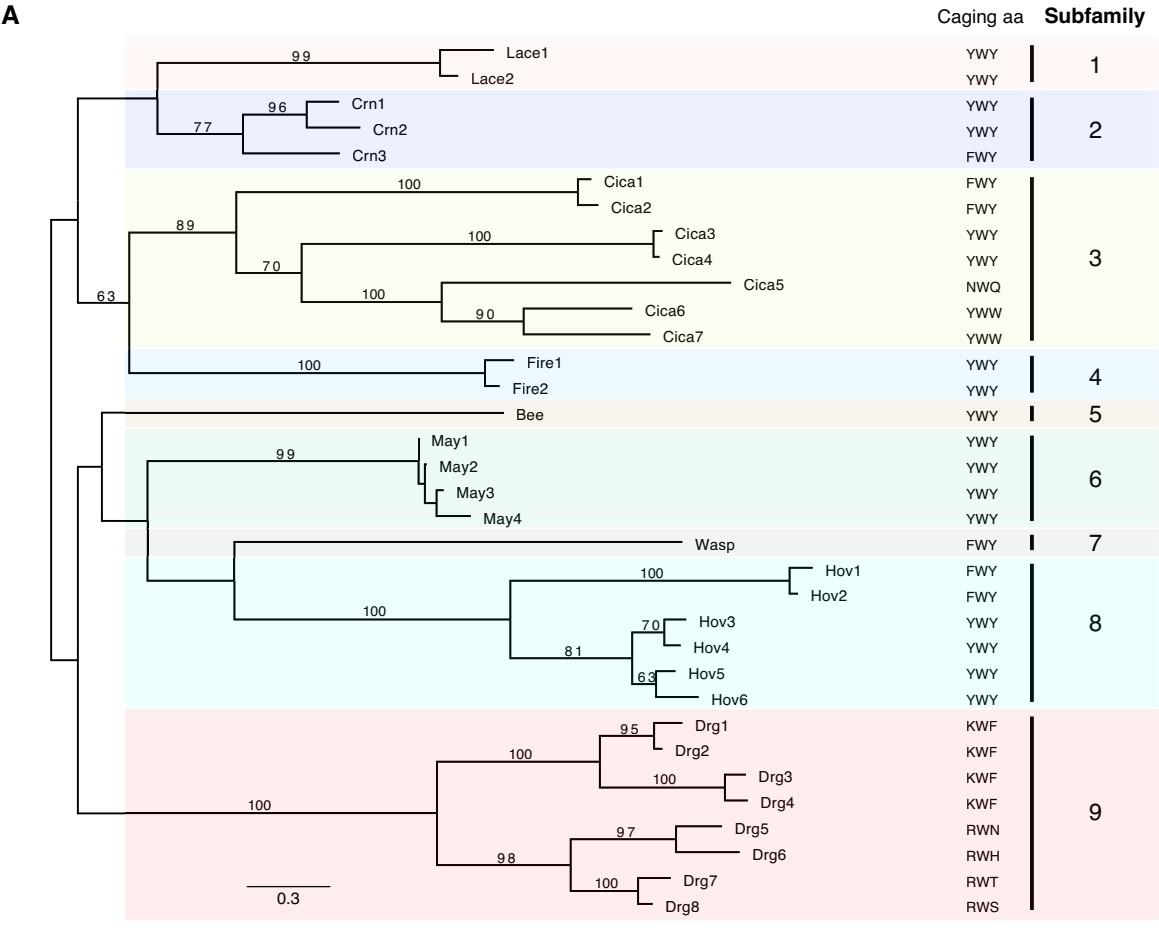

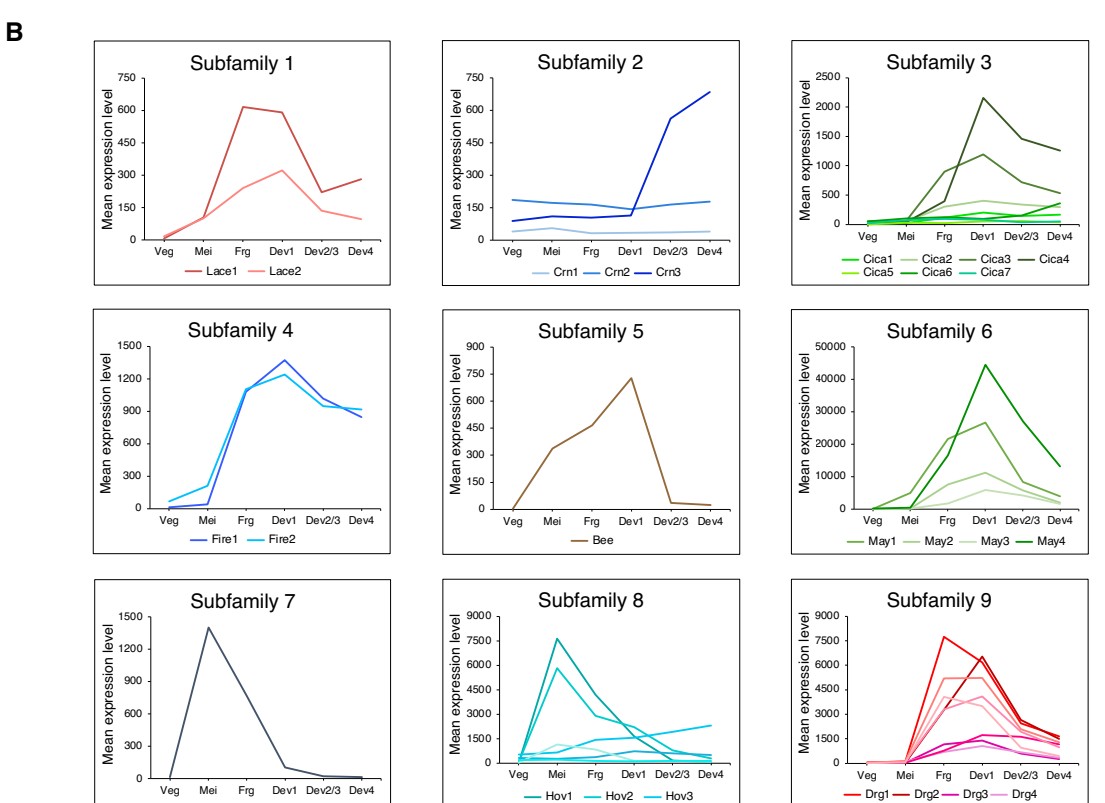

Figure 1. Identification of *P. tetraurelia* chromodomain proteins.

(A) ML phylogenetic tree reconstructed from full-length chromodomain protein sequences. Bootstrap supports greater than 50 are shown. Proteins were clustered into nine subfamilies based on the phylogenetic relationship and branch support. Caging aa denotes the amino acids located in the positions of the aromatic cage.
(B) Expression profiles of each gene in each subfamily, generated with published DESeq2-normalized RNA-seq counts retrieved from the *Paramecium* Database (Arnaiz et al, 2020; Arnaiz et al, 2017). The horizontal axis denotes developmental stages from early to late stage of autogamy, as defined in the *Paramecium* Database. Veg vegetative, Mei MIC meiosis, Frg maternal MAC fragmentation, Dev1 early new MAC development, Dev2/3 intermediate new MAC development, Dev4 late new MAC development. See also Appendix Fig. S1.

chromodomain proteins in this organism may reflect the extensive genome rearrangements occurring during sexual development, which necessitates additional organizational measures to effectively manage such complex genetic alterations.

We further examined the phylogenetic relationship of these proteins, which could be divided into 9 subfamilies (Fig. 1A,B). The protein sequences were mainly aligned at their CDs, the most conserved part of the proteins, and a phylogenetic tree generated from the CDs only had a negligible effect on the tree topology (Appendix Fig. S1). Since we were searching for putative readers of H3K9me3 and H3K27me3, we next investigated whether these candidates possessed the three aromatic residues required for binding methylated lysines (Jacobs and Khorasanizadeh, 2002; Nielsen et al, 2002). Alignment of the CDs revealed that all members of Subfamily 9 had divergent aromatic residues, either in the first residue or both the first and the last residues (Fig. 1A; Appendix Table S2). Although all of them belonged to the "intermediate cluster" and were development-specific, we did not examine these further since the absence of an intact aromatic cage suggests that these do not bind methylated lysines (Fig. 1B). Two other subfamilies, Subfamily 2 and 8, as well as lowly expressed members of Subfamily 3 (<500 mean expression level) were also excluded, since they contained predominantly vegetatively expressed genes, had very low expression levels overall or divergent aromatic residues. The remaining candidates contained an intact aromatic cage and were exclusively expressed during sexual development. We therefore went on to perform a silencing screen of these candidates, with a readout of progeny survival assessed by survival tests and DNA elimination defects assessed by IES retention PCRs. Survival tests are performed to assess whether the new MAC is functional after autogamy and is based on the division rate or cell death of the silencing culture compared to the control silencing; IES retention PCRs are performed on genomic DNA extracted after autogamy using primers flanking IESs to assess IES retention. From these screens, only two subfamilies (Subfamily 4: Firefly1, Firefly2; Subfamily 6: Mayfly1, Mayfly2, Mayfly3, Mayfly4) were essential for post-development survival and DNA elimination (Fig. 2A–E).

## Fire1/2 and May1-4 have distinct localization patterns in developing new MACs

Our small-scale silencing screen identified two subfamilies of chromodomain proteins essential for survival and DNA elimination. The first (Subfamily 4) consists of Firefly1 (Fire1, PTET.51.1.G0220149) and Firefly2 (Fire2, PTET.51.1.G0020347), a pair of ohnologues that contain one chromodomain at the N terminus and a long, disordered C terminus (Fig. 2A). Their expression is development-specific and peaks at early new MAC

development (Dev1) (Fig. 2B) (Arnaiz et al, 2017). To determine the subcellular localization of the proteins, we transformed cells with GFP-tagged Fire1 and Fire2 and observed them throughout autogamy (sexual process involving self-fertilization). The proteins are mainly detected in the early developing new macronuclei (MACs), and fade as development progresses and the new MACs enlarge (Fig. 2C; Appendix Fig. S2). There was also a very faint signal in the maternal MAC at early stages of development, but much weaker than the new MAC signal and often difficult to detect. As the proteins shine bright in the developing new MACs only for a short time window before fading, the proteins were named Firefly (Fire).

The second subfamily, Subfamily 6, consists of four closely related ohnologues: Mayfly1 (May1, PTET.51.1.G1380122), Mayfly2 (May2, PTET.51.1.G0800198), Mayfly3 (May3, PTET.51.1.G1450044) and Mayfly4 (May4, PTET.51.1.G1520043) (Fig. 2A). These were the most highly expressed genes in our study, peaking at the late stage of development and reaching a maximum expression as high as the scnRNA-binding PIWI protein Ptiwi09 (Figs. 1B and 2B). In our initial domain search, all May proteins contained a CD, but May1 also had a chromo shadow domain (CSD) characteristic of HP1-family chromodomain proteins. A subsequent Phyre2 search identified putative CSDs in May2 and May3 as well (Kelley et al, 2015). We next investigated their subcellular localization by transforming cells with a GFP-tagged May1. The May1 protein is found exclusively in the developing new MACs in the late stage of development, where it localizes to distinct compartments and form nuclear foci (Fig. 2D). These foci appear dynamic and display variable sizes that at first are small, then increasing in size and decreasing in number as development progresses (Appendix Fig. S3). HP1-family proteins such as HP1a in *Drosophila* and HP1α in humans have been shown to phase separate into liquid droplets, a feature required to form heterochromatin domains (Larson et al, 2017; Strom et al, 2017). Hence, the localization of May proteins into distinct, dynamic foci may be indicative of phase separation. Compartmentalization into foci may facilitate biological processes that require high concentrations of effectors or bring together distant DNA sequences. Of note, the excisase Pgm also localizes to distinct compartments, suggesting that IESs may be spatially concentrated prior to excision (Baudry et al, 2009). In *Tetrahymena*, a ciliate performing similar DNA elimination processes as *Paramecium*, the HP1-family chromodomain protein Pdd1 is required for IES elimination and forms nuclear foci as well (Kataoka and Mochizuki, 2015; Kataoka et al, 2016; Madireddi et al, 1996). The formation of such condensates appears to be a pre-requisite for IES elimination in this organism. However, whether May proteins indeed phase separate, and whether this is required to initiate or facilitate compartmentalization of other effectors required for IES excision remains to be

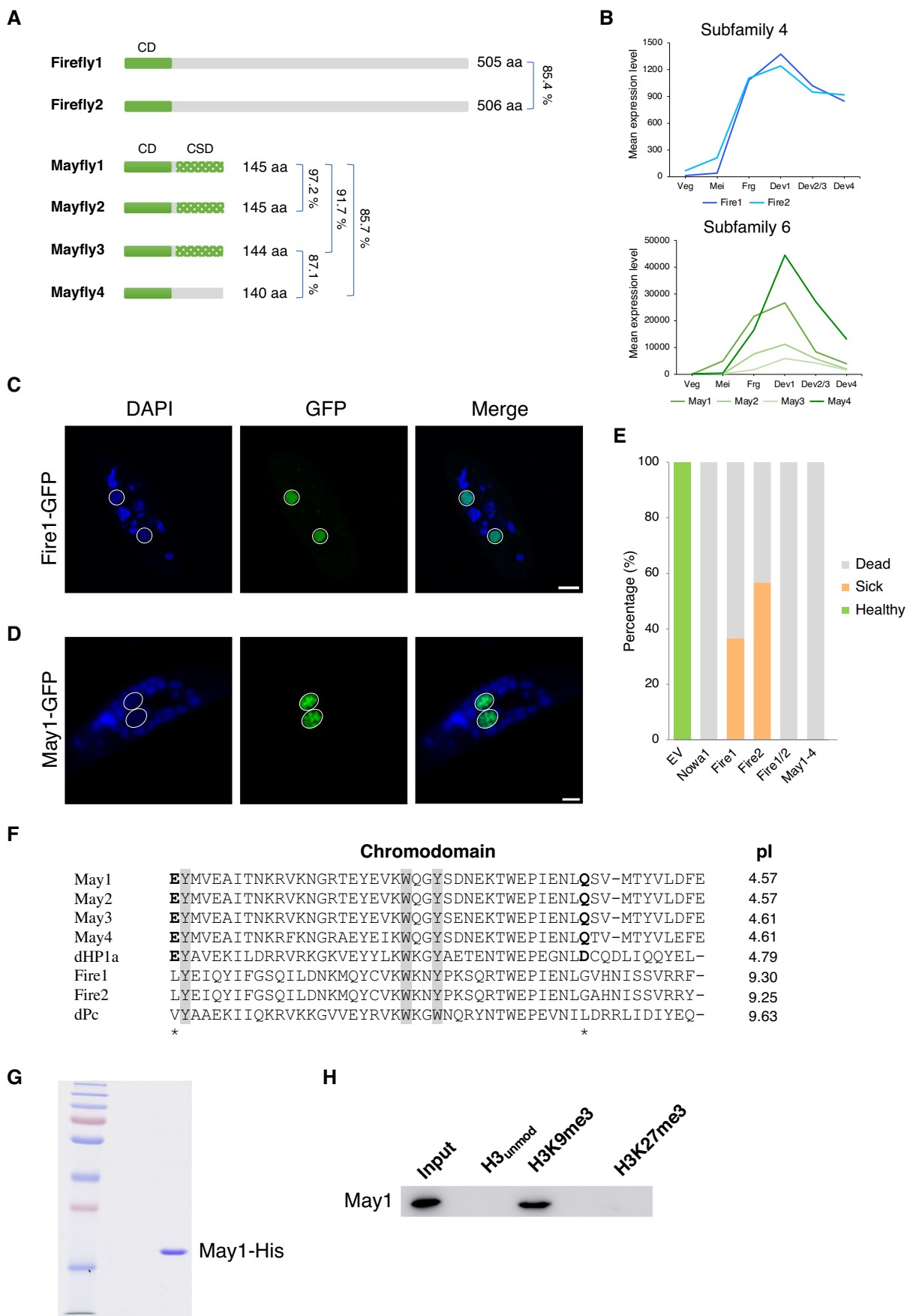

Figure 2. May1-4 recognize H3K9me3, and Fire1/2 are putative readers of H3K27me3.

(A) Domain organization, size, and identity between ohnologues of Fire and May proteins. CD: chromodomain, CSD: chromo shadow domain. (B) Expression profiles of each gene in each subfamily, generated with published DESeq2-normalized RNA-seq counts retrieved from the *Paramecium* Database (Arnaiz et al, 2020; Arnaiz et al, 2017). The horizontal axis denotes developmental stages from early to late stage of autogamy, as defined in the *Paramecium* Database. Veg vegetative, Mei MIC meiosis, Frg maternal MAC fragmentation, Dev1 early new MAC development, Dev2/3 intermediate new MAC development, Dev4 late new MAC development. (C, D) Localization of Fire1-GFP (C) and May1-GFP (D) in the late stage of development. Dotted white circles denote new MACs. Scale bar: 10 μm. (E) Survival test after silencing of Fire and May, using EV as a negative control and Nowa1 as a positive control. n = 30, number of cells examined. Green: Healthy, Orange: Sick, Gray: Dead. (F) Chromodomain alignment using MUSCLE (v3.8) (Edgar, 2004). Aromatic cage residues are shaded. The asterisk denotes the clasp residues outside of the aromatic cage, and the predicted isoelectric point (pI) is listed on the right. Polar clasp residues are marked in bold. *Drosophila* Pc (dPc) and HP1a (dHP1a) are included as references. (G) Coomassie staining of 1 μg of recombinant May1-His purified from *E. coli*. (H) Western blot of peptide pull-down assays using purified recombinant May1-His from *E. coli* and the commercially available histone peptides listed in Appendix Table S4. See also Appendix Figs. S2–S4. Source data are available online for this figure.

determined. To follow a similar naming convention as Fire1/2, we named these proteins Mayfly1-4 (May1-4). Subsequently, the remaining chromodomain proteins were also named following this convention.

## May1-4 recognize H3K9me3, and Fire1/2 are putative readers of H3K27me3

In most organisms, H3K27me3 and H3K9me3 are set by different machineries and perform different functions: H3K27me3 is important for developmental gene repression and H3K9me3 for transposon control. In *Paramecium*, however, the PRC2 complex was shown to have dual methyltransferase activity and sets H3K27me3 as well as H3K9me3 during development (Frapporti et al, 2019; Lhuillier-Akakpo et al, 2014; Miro-Pina et al, 2022; Wang et al, 2022). Both methylation marks have been reported to co-occur on the same TE elements in *Paramecium*, but their role in IES elimination remains unclear. Moreover, the only studies investigating their function focus on the Ezl1 methyltransferase or the complex it belongs to, making it impossible to assign separate functions to each of these marks. The identification of two subfamilies of chromodomain proteins required for this process may therefore allow us to interrogate the function of each of these marks separately if they recognize different marks. We therefore sought to investigate their binding preferences. Several well-characterized proteins contain CDs which specifically recognize H3K9me3 or H3K27me3, such as the *Drosophila* Heterochromatin protein 1 (dHP1) and Polycomb (dPc) proteins (Bannister et al, 2001; Fischle et al, 2003; Lachner et al, 2001; Min et al, 2003). In addition to the aromatic cage residues required for binding methylated lysines, two other key factors have been identified as determinants for discriminating between H3K9me and H3K27me: the polarity of two residues outside of the aromatic cage (clasp residues), as well as the acidity of the CD (Kaustov et al, 2011). The two clasp residues form a "polar clasp" and the CD is acidic in HP1-family members that recognize H3K9me, whereas they form a "hydrophobic clasp" and the CD is more basic in Pc-family members that recognize H3K27me. Alignment of the CDs of Fire and May with *Drosophila* dHP1 and dPc CDs revealed that both Fire and May possess the three aromatic residues required for binding to methylated lysines (Fig. 2F; Appendix Table S2) (Jacobs and Khorasanizadeh, 2002; Nielsen et al, 2002). Moreover, the CDs of Fire1/2 have nonpolar residues in both positions and a corresponding basic CD, indicating that these proteins may be H3K27me3 readers. On the other hand, May1-4 have polar residues in both positions and a corresponding acidic CD, suggestive of a

preference for H3K9me3. This is also consistent with the presence of both a CD and a CSD in May proteins. Strikingly, when investigating the CDs of all chromodomain proteins we identified in *Paramecium*, only Fire1/2 contained intact aromatic residues, two nonpolar clasp residues and a corresponding basic pI allowing a prediction of putative binding preferences for H3K27me.

We hypothesized that if May1 binds one of the methylation marks the PRC2 complex sets as predicted from the alignments, the localization of May1 into nuclear foci may be used as a readout for binding. If the foci assemble after binding to the methylation mark, the absence of the PRC2 complex should alter its localization. Therefore, we investigated the localization of May1-GFP in the absence of PtCAF1, having previously shown that the PRC2 complex is not stable in the absence of this subunit (Wang et al, 2022). As expected, the localization of May1-GFP is altered, with a few bright foci remaining and an overall more homogenous distribution than in the wild type (Appendix Fig. S4). The remaining foci may either be due to the presence of residual complexes as we can only perform knockdown and not knockout in *Paramecium*; or alternatively, a part of the recruitment of May1 to chromatin is PRC2-independent. While these results suggest that the correct localization of May1 depends on the PRC2 complex, this assay cannot exclude indirect effects or distinguish between the two marks the PRC2 complex sets. Therefore, to directly determine the binding preference of May proteins, we re-codonized the May1 gene sequence for protein expression and purification in *E. coli* (Fig. 2G; Appendix Table S3). Recombinant, purified May1-His was then used for in vitro peptide pull-down assays using biotinylated histone peptides to investigate its binding preference (Appendix Table S4). As predicted from the alignments, we confirmed that May1 binds H3K9me3, but not H3K27me3 (Fig. 2H).

We also re-codonized the Fire1 gene sequence for protein expression and purification in *E. coli* to assess its binding preference as we did for May1; however, this unfortunately failed due to solubility issues. This is likely because Fire proteins are highly disordered. Since readers can associate to writers, we next performed immunofluorescence to assess the abundance of H3K9me3 and H3K27me3 in the EV control and Fire1/2-silenced culture. Interestingly, the signal of both modifications disappeared from the new MACs when Fire1/2 was silenced, whereas this did not affect the H3K27me3 signal in the maternal MAC (Fig. EV1). This may either suggest that Fire1/2 binds to H3K27me3 in the new MACs but is required for the deposition of H3K9me3, or that Fire1/2 binds to both H3K9me3 and H3K27me3. We favor the first hypothesis, since Fire1/2 contain a Pc-like CD. However, it is also possible that Fire1/2 have dual binding preferences for both

H3K9me3 and H3K27me3. We conclude that Fire1/2 are putative readers of H3K27me3 but affect the deposition of both H3K9me3 and H3K27me3 in the new MACs.

## Fire and May proteins do not fulfill the same role in DNA elimination

The localization patterns of Fire and May proteins suggest that they may perform different functions in the DNA elimination process, despite being dependent on the same methyltransferase. To get a comprehensive view of the IES retention after Fire1/2 and May1-4 silencing, we deep-sequenced genomic DNA and analyzed genome-wide IES retention scores (IRS). IRS values were calculated for each of the 45,000 IESs. For each IES, an IRS from 0 to 1 is a measurement of 0 to 100% retention in the sequencing, i.e., how many reads include vs. does not include the IES. This means that there is a gradient in IES retention after depletion of key proteins involved in their elimination rather than complete retention or excision. Moreover, some IESs can be retained very weakly due to stochastic reasons, which typically only occurs in some copies of the genome (the ploidy of which is 800n) or in very few cells. To exclude noise coming from weakly retained IESs and such stochastic events, we chose a cutoff of IRS > 0.1 to denote retained IESs in the following analyses. The depletion of Fire1/2 affected 14,684 IESs (IRS > 0.1), whereas May1-4 silencing affected a smaller subset of 8616 IESs (IRS > 0.1) (Fig. 3A,B). The IRS distribution of May1-4 silencing resembles the bimodal retention pattern after Ezl1 depletion with many IESs weakly affected, as well as a second peak of IESs that are more strongly affected by the silencing (Lhuillier-Akakpo et al, 2014). Both Fire1/2 and May1-4-dependent IESs fall within the Ezl1-dependent subset and share a considerable overlap (Figs. 3B and EV2).

Examining the size distribution of each subset revealed a striking difference between the effect of Fire1/2 and May1-4 depletion (Fig. 3C; Appendix Figs. S5 and S6). While Fire1/2 is required for the excision of IESs as short as 26 bp (35.7% of Fire1/2-dependent IESs are <50 bp), May1-4 depletion predominantly affects larger IESs and only 3.4% of the retained IESs are shorter than 50 bp (Fig. 3C). About half of all IESs in *Paramecium* are shorter than 50 bp (49.7%), and of these, only 1.3% are retained after May1-4 depletion. Since May1-4 and Fire1/2 are chromodomain proteins and IESs are more likely to be covered by one or more nucleosomes when they are longer, we examined the retention of short (<200 bp) and long (>200 bp) IESs separately. However, since there are far fewer long IESs than short IESs, we examined the retained IES subsets in relation to Pgm-KD, which denotes all IESs (Appendix Figs. S5 and S6). While May1-4-KD only affects 17.2% of IESs under 200 bp, they are required for the elimination of 64.8% of IESs over 200 bp. Fire1/2-KD affects a similar percentage of large IESs (66.3%), but nearly double the number of short IESs (31.3%). Since the loss of Fire1/2 affected the deposition of both H3K9me3 and H3K27me3 in the new MACs, and we noted an overlap in IESs dependent on Fire1/2 and May1-4, we next investigated whether Fire and May proteins are partly redundant through binding H3K9me3. If Fire and May proteins both recognize H3K9me3, double silencing of Fire1/2 and May1-4 (Fire/May-KD) should increase the number of retained IESs beyond the combined subset of the single silencings, as some IESs may be affected by neither Fire1/2 nor May1-4 single silencings due to redundancy. However,

this was not the case (Fig. EV3). We therefore hypothesize that Fire1/2 binds H3K27me3, which affects the downstream deposition of H3K9me3 and therefore the function of May1-4. In sum, we conclude that May proteins are predominantly required for the excision of longer IESs, and that Fire and May proteins affect different subsets of PRC2-dependent IESs.

## Fire1/2 act in tight association with TFIIS4

Our small-scale silencing screen indicated that the IES retention pattern after Fire1/2 depletion mimics a previously described protein, the transcription elongation factor TFIIS4 (Maliszewska-Olejniczak et al, 2015). TFIIS4 was previously demonstrated to be involved in noncoding RNA transcription in the developing new MACs, which are believed to act as scaffolds for Piwi-bound sRNAs to target IESs for elimination. A putative link between Fire1/2 and TFIIS4 was also strengthened when examining the expression profile of TFIIS4 as well as its localization pattern, both of which resembled our observations of Fire1/2 (Appendix Fig. S7A) (Maliszewska-Olejniczak et al, 2015). A correlation plot of genome-wide IES retention phenotypes upon silencing of various chromatin-related factors involved in DNA elimination confirmed that the IES subsets affected by Fire1/2 and TFIIS4 correlate very well, and the two datasets share a near-complete overlap in IES retention when considering all retained IESs (IRS > 0) (Figs. 3D, 4A–C; Appendix Fig. S7B). May1-4 dependent IESs, however, did not correlate very well with any other silencing, and appears to be a distinct subset of PRC2-dependent IESs (Figs. 3D and EV2B).

The strong correlation in IRS after TFIIS4 and Fire1/2 depletion suggests that these proteins may act in the same pathway. If this is the case, double knockdown of Fire1/2 and TFIIS4 (Fire/TFIIS4-KD) should mimic the single knockdowns. In agreement with this hypothesis, the IES retention after Fire/TFIIS4-KD correlated very strongly with Fire1/2 and TFIIS4-KD, and did not affect many additional IESs (Fig. 4C). We next tested what happens to Fire1 and TFIIS4 in the absence of the other. To determine this, we first silenced Fire1/2 in cells expressing TFIIS4-GFP. After Fire1/2 depletion, no GFP signal could be observed in the new MACs, and the reciprocal experiment yielded similar results (Fig. 4D). Northern blots confirmed that this was not at the level of transcription, suggesting that Fire1/2 and TFIIS4 act in tight association with each other at the protein level (Fig. 4E). While these analyses do not allow us to conclude whether the proteins are simply mis-localized to the cytoplasm or absent, they need each other for correct nuclear localization and function. We conclude that Fire1/2 act in tight association with TFIIS4 in the nucleus, which may suggest a functional link between noncoding RNA transcription and histone modifications in IES elimination.

## Fire1/2 are dispensable for scnRNA biogenesis but required for the accumulation of iesRNAs

Our findings suggest that IESs are retained in Fire1/2 silencing due to the inhibition of noncoding RNA transcription used as scaffolds for targeting by sRNAs. In favor of this hypothesis, we observed a strong correlation in IES retention between Fire1/2 silencing and Dcl2/3/5 silencing (Figs. 3D and 4C,F). Dcl2/3/5 are Dicer-like proteins that generate two classes of sRNAs required for IES elimination: scan RNAs (scnRNAs) and iesRNAs (Lepere et al,

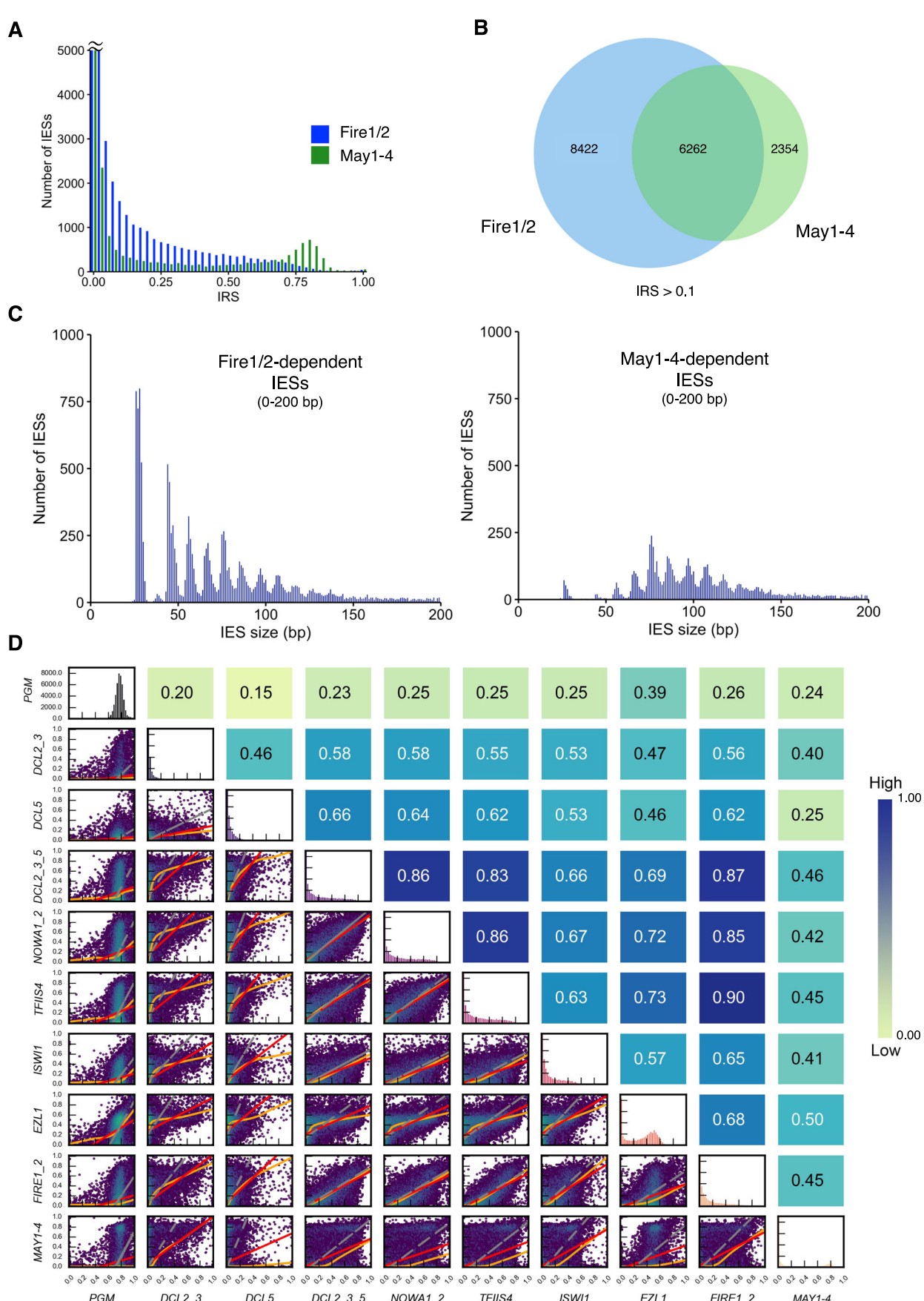

**Figure 3. Fire and May proteins do not fulfill the same role in DNA elimination.**

(A) IES retention score distributions after Fire1/2 or May1-4 silencing, for IESs with an IRS > 0.1. Histogram bars with more than 5000 IESs have been truncated for clarity. (B) Venn diagram depicting shared IES retention between Fire1/2 and May1-4 silencing, for IESs with an IRS > 0.1. (C) Size distribution of retained IESs (IRS > 0.1) after Fire1/2 (left) or May1-4 (right) depletion. Note that the x-axis has been cropped to only show IESs with a size of 0–200 bp. Size distribution of retained IESs larger than 200 bp are found in Appendix Fig. S5. (D) Correlation plots calculated by hexagonal binning of IES retention scores generated using After_ParTIES (Swart et al, 2017) and the IES retention scores provided in Dataset EV2. Pearson's correlation coefficients are given above each subgraph. Red lines are for ordinary least-squares (OLS) regression, orange lines for LOWESS, and gray lines for orthogonal distance regression (ODR). From light green to dark blue, the correlation is stronger. See also Figs. EV2, EV3, Appendix Figs. S5 and S6. Source data are available online for this figure.

2009; Sandoval et al, 2014). Dcl2/3/5-depletion therefore abolishes both classes of sRNAs, suggesting that TFIIS4 may be required for sRNA targeting. However, we would also observe the same effect if Fire1/2 and TFIIS4-dependent transcription was required for generating the precursors of these sRNAs. To investigate this, we deep-sequenced sRNAs extracted from cultures depleted of Fire1/2 or a control silencing and examined their sequence content (Fig. 4G). Three classes of sRNAs can be distinguished in the control culture: 23 nt endo-siRNAs produced by Dcr1, 25 nt scnRNAs produced by Dcl2/3, as well as 26-31 nt iesRNAs that are Dcl5 dependent and match exclusively to IES sequences (Lepere et al, 2009; Sandoval et al, 2014; Solberg et al, 2023). ScnRNAs are generated by bidirectional transcription of the MIC genome at the onset of autogamy and are produced from both MAC and IES sequences. Since MAC sequences outnumber IES sequences, MAC-matching scnRNAs initially outnumber IES-matching scnRNAs in the early stage of development. By the late stage, scnRNAs are predominantly matching to IES sequences and other eliminated sequences as a consequence of the selective degradation of MAC-matching scnRNAs during the scanning process. Therefore, we can use this ratio to assess whether scanning has occurred. The ratio of IES-matching to MAC-matching scnRNAs indicates that scanning occurred in all cultures and the scnRNA population appears largely unaffected in both samples (Fig. 4G). We conclude that Dcl2/3-dependent IESs are not retained upon Fire1/2 silencing due to an indirect effect on the population of scnRNAs, nor due to an effect on the scanning process.

In contrast, the iesRNA population is greatly reduced in the Fire1/2-silenced sample (Fig. 4G). The iesRNA pathway is a feedback loop initiated by IES excision to ensure the removal of all copies of IESs. In this pathway, excised IESs concatenate, circularize, are bidirectionally transcribed and cleaved to generate iesRNAs (Allen et al, 2017). Because the production of iesRNAs requires excised IESs, the loss of iesRNAs in the Fire1/2-silenced sample may reflect the retention of a large fraction of the total IES population which then cannot generate iesRNAs. Alternatively, Fire1/2 may be involved in iesRNA biogenesis and transcription of IES concatemers. To investigate this, we calculated the contribution of Fire1/2-dependent and independent IESs to the iesRNA pool. This analysis revealed that although Fire1/2 only affects the excision of about a third of the IESs, the contribution of these IESs to the total iesRNA pool is greater, with 54.1–80.5% of iesRNAs being Fire1/2-dependent (IRS > 0.1 or IRS > 0.01, respectively) (Fig. EV4A). This is likely due to the size of the IESs that are retained in the Fire1/2-silenced culture, as we find most long IESs to depend on Fire1/2, and these IESs have a greater count of iesRNAs per IES than Fire1/2-independent IESs (Fig. EV4B). We

next assessed whether Fire1/2 silencing also affects Fire1/2-independent iesRNAs to determine if Fire1/2 may be involved in the transcription of concatemers, or if the reduction in iesRNAs is purely a consequence of the IES retention (Fig. EV4B). This analysis revealed a much greater effect on Fire1/2-dependent than independent iesRNAs between the EV and Fire1/2 silencing, which may suggest that Fire1/2 are not involved in the transcription of IES concatemers. However, there is also a slight reduction of Fire1/2-independent iesRNAs when only considering IESs with an IRS > 0.1 (Fig. EV4B, left), but this may be a consequence of weakly retained Fire1/2-dependent IESs as this reduction is not observed when considering IESs with an IRS > 0.01 (Fig. EV4B, right). We conclude that Fire1/2 are required for the accumulation of iesRNAs, likely through facilitating IES excision, but are not required for the biogenesis of scnRNAs, nor the scanning process.

## Fire1/2 are required for ISWI1-mediated nucleosome depletion

We recently reported that IESs are nucleosome-depleted prior to their excision, a feature dependent on sRNAs and the nucleosome remodeler ISWI1 (Singh et al, 2022). If sRNAs are needed to guide ISWI1 and targeting by sRNAs requires noncoding RNAs transcribed in a TFIIS4-dependent manner, the absence of Fire1/2 should prevent nucleosome depletion of sRNA-dependent IESs. To test this hypothesis, we performed nucleosome profiling in the absence of Fire1/2. Cells were depleted of both Fire1/2 and PiggyMac (Pgm), the excisase responsible for IES excision (Baudry et al, 2009). As a control, we used an empty L4440 vector, EV control, producing siRNAs matching the plasmid, which was also co-silenced with Pgm. Pgm was co-silenced to retain all IESs which allows us to perform analyses also on Fire1/2-independent IESs that would otherwise be removed during the course of development. From these cultures, we isolated macronuclear DNA to calculate IES retention scores and nucleosomal DNA to assess nucleosome occupancy over IESs. As IESs are underrepresented in these samples, we used the macronuclear DNA dataset to assess the input over IESs when calculating nucleosome occupancies. Examining nucleosome occupancy over all IESs and the sRNA-dependent IES subset, revealed markedly higher nucleosome densities in the latter (Appendix Fig. S8). However, this may also reflect the sizes of IESs that are dependent on sRNAs, as they tend to be larger and can therefore contain more nucleosomes. When analyzing nucleosome densities between EV/Pgm and Fire1/2/Pgm over all IESs, we did not find any obvious differences (Fig. 5C; Appendix Fig. S8). This is to be expected, as Fire1/2 silencing does not affect the excision of all IESs. On the other hand, a modest increase in nucleosome density

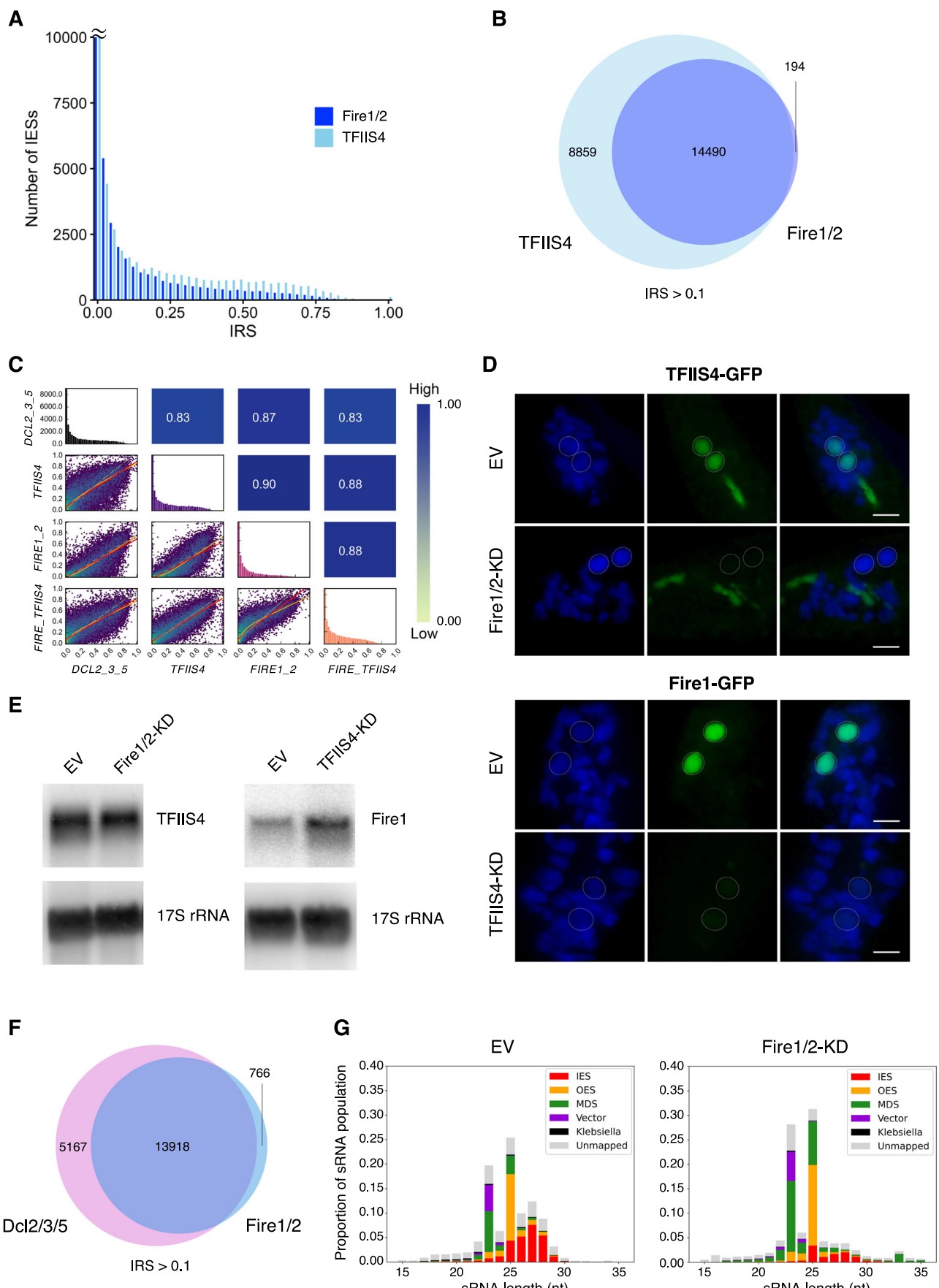

◀

**Figure 4. Fire1/2 act in tight association with TFIIS4.**

(A) IES retention score distribution after Fire1/2 or TFIIS4 silencing, for IESs with an IRS > 0.1. Histogram bars with more than 10,000 IESs have been truncated for clarity. Note that the cutoff is different from Fig. 3A. (B) Venn diagram depicting shared IES retention between TFIIS4 and Fire1/2 silencing, for IESs with an IRS > 0.1. (C) Correlation plots calculated by hexagonal binning of IES retention scores generated using After_ParTIES (Swart et al, 2017) and the IES retention scores provided in Dataset EV2. Pearson's correlation coefficients are given above each subgraph. Red lines are for ordinary least-squares (OLS) regression, orange lines for LOWESS, and gray lines for orthogonal distance regression (ODR). From light green to dark blue, the correlation is stronger. (D) Localization of TFIIS4-GFP with and without Fire1/2, as well as Fire1-GFP with and without TFIIS4. Dotted white circles denote new MACs. Scale bar: 10 μm. (E) Northern blots using RNA extracted from the late stage of development in EV, TFIIS4 silencing and Fire1/2 silencing cultures, probed against TFIIS4, Fire1, or the 17S rRNA as a loading control. (F) Venn diagram depicting shared IES retention between Dcl2/3/5 and Fire1/2 silencing, for IESs with an IRS > 0.1. (G) Analysis of sRNA sizes and contents (color coded) of sRNAs extracted from EV control and Fire1/2-silenced samples at the Late stage of development, plotted as a proportion of the total sRNA population. IES internally eliminated sequence, OES other eliminated sequence, MDS macronuclear destined sequence (MAC matching), TE transposable element, Vector pGEM T and L4440 vectors, Klebsiella *Klebsiella pneumoniae* (food source for *Paramecium*). See also Fig. EV4 and Appendix Fig. S7. Source data are available online for this figure.

upon Fire1/2/Pgm silencing is observed when only considering sRNA-dependent IES subsets, such as those dependent on Dcl2/3, Dcl5, and Dcl2/3/5, which also comprise the Fire1/2-dependent IESs. Since the majority of IESs are shorter than the size of a nucleosome (93% are <150 bp) and thus may be less likely to be covered by a nucleosome, we wished to analyze longer IESs in particular (Arnaiz et al, 2012). The excision of long IESs depends on Fire1/2 and sRNAs to a greater extent than short IESs, which may be due to higher nucleosome occupancy and a need for nucleosome remodeling to facilitate DNA elimination, as previously hypothesized (Appendix Figs. S5 and S6) (Singh et al, 2022). Corroborating this finding, we observed a stronger effect on longer IESs (>200 bp), in which higher nucleosome densities were observed in the Fire1/2/Pgm sample than in the control (Appendix Fig. S8).

We noted a considerable overlap of the IES subsets affected by the depletion of Fire1/2 and ISWI1, a chromatin remodeler required for nucleosome depletion of IESs (Fig. 5A,B) (Singh et al, 2022). Therefore, we compared our data to the previously published nucleosome profiling data upon ISWI1 silencing to determine if their effects on the nucleosomal landscape are similar. Comparison with ISWI1 revealed that the phenotype of Fire1/2 depletion on nucleosome densities was comparable, if not stronger, than the effect of ISWI1 depletion (Fig. 5C–E). For ISWI1-dependent IESs, both short and long IESs had significantly higher nucleosome densities in Fire1/2 and ISWI1 silencing than the control (Fig. 5E). However, this was not the case for ISWI1-independent IESs, which appeared to follow the opposite trend. These findings suggest that Fire1/2 proteins are required for driving ISWI1-dependent nucleosome depletion on IESs, and that this effect is specific to ISWI1-dependent IESs. In addition, our results suggest that open chromatin may not be a pre-requisite for transcription but a consequence thereof, which is consistent with transcription initiation being mediated through histone modifications rather than DNA sequences.

Our results allow us to propose a model for the role of PRC2 on IES elimination by connecting it to noncoding RNA transcription and ISWI1-mediated nucleosome depletion. Based on our findings, we hypothesize that the PRC2 complex sets H3K9me3 and H3K27me3 during development, one or both of which recruits Fire1/2 and TFIIS4 through the chromodomain of Fire1/2 (Fig. 6). This leads to the recruitment of RNA polymerase II and noncoding RNA transcription, the transcripts of which serve as scaffolds for targeting by sRNAs. The sRNAs recruit the nucleosome remodeler

ISWI1, resulting in nucleosome depletion on IESs which facilitates their recognition by the excision machinery. Finally, the IES is removed by the domesticated transposase Pgm, and the broken ends are re-ligated by the non-homologous end-joining machinery.

## Discussion

Our group and others recently reported the characterization of the *Paramecium* Polycomb repressive complex 2 (PRC2) and its involvement in TE and IES elimination (Miro-Pina et al, 2022; Wang et al, 2022). Moreover, the catalytic subunit of the PRC2 complex, Ezl1, has recently been shown to set both H3K9me3 and H3K27me3 on TEs during development, which is required for transcriptional repression and subsequent removal of the sequences (Frapporti et al, 2019; Lhuillier-Akakpo et al, 2014). However, the requirement of this complex for the precise elimination of IESs much shorter than the size of a nucleosome remained enigmatic. Here, we clarified the role of these methylation marks for the removal of IESs through the identification and characterization of two subfamilies of chromodomain proteins, Fire1/2 and May1-4. Our results suggest that May1-4 and Fire1/2 are not redundant, but instead perform different functions to facilitate IES elimination. We demonstrated that the involvement of Fire1/2 provides a mechanistic link connecting the PRC2 complex to TFIIS4-dependent ncRNA transcription in the developing new MACs, which are then used as scaffolds for targeting by sRNAs followed by IES elimination. These findings suggest that the involvement of the PRC2 complex in DNA elimination differs between TEs and IESs. For TEs, it induces heterochromatin formation and transcriptional repression, whereas for IESs, it promotes transcription and nucleosome depletion.

The retention of IESs shorter than the size of a nucleosome upon PRC2 depletion has long been a mystery. It was suggested that H3K27me3 and H3K9me3, the two methylation marks the complex sets, may be placed locally on nucleosomes that overlap IESs (Lhuillier-Akakpo et al, 2014). This model was heavily based on findings from the related ciliate *Tetrahymena*, which has considerably longer IESs than *Paramecium* (Arnaiz et al, 2012; Hamilton et al, 2016). In this ciliate, H3K9me3 and H3K27me3 were shown to physically interact with IESs, and the heterochromatinization of IESs into heterochromatin bodies appears to be a pre-requisite for IES elimination (Kataoka and Mochizuki, 2015; Kataoka et al, 2016; Liu et al, 2007; Taverna et al, 2002). While this model appears to

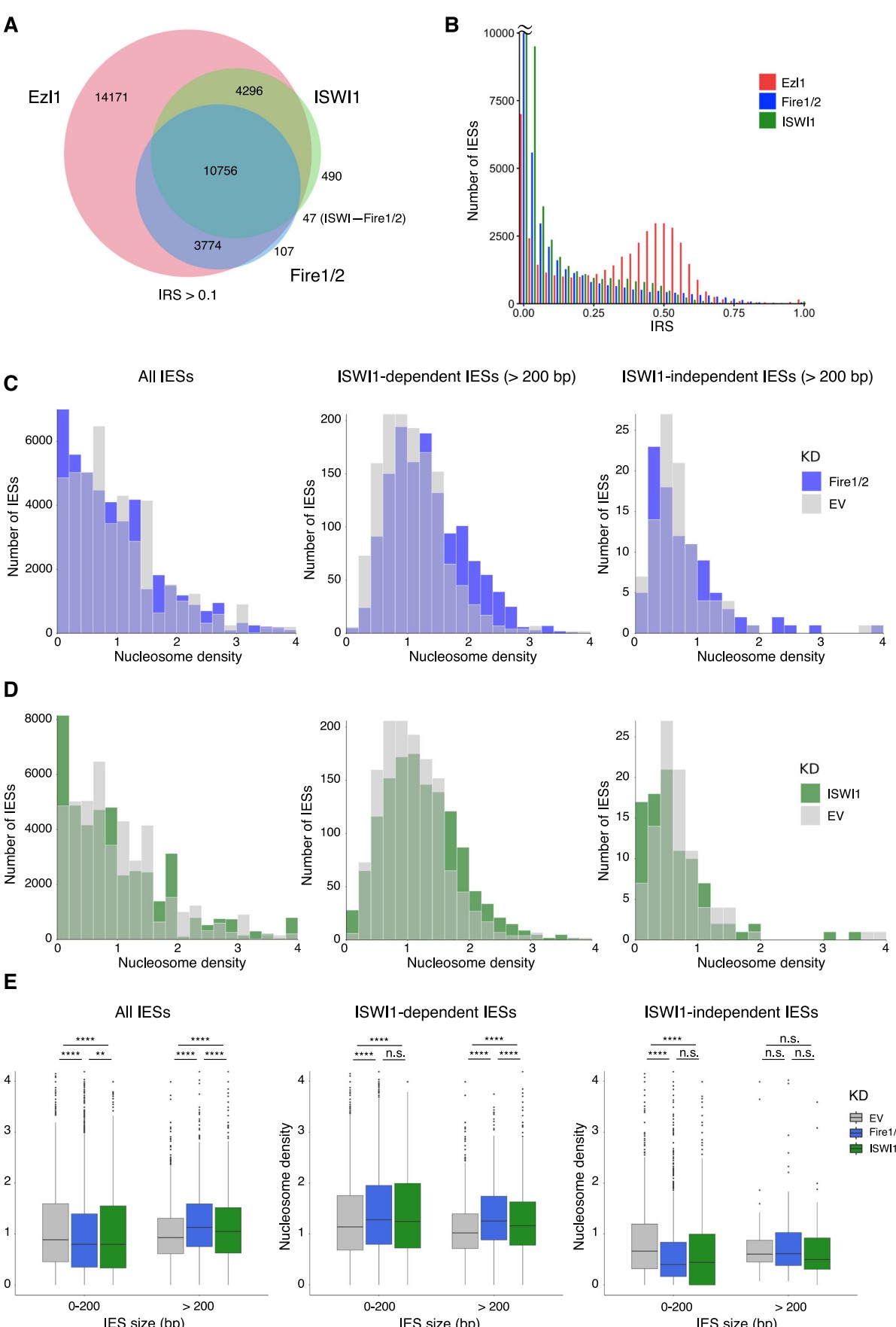

**Figure 5.  Fire1/2 proteins are required for ISWI1-mediated nucleosome depletion.**

(A) Venn diagram depicting shared IES retention between Fire1/2, Ezl1, and ISWI1 silencing, for IESs with an IRS > 0.1. (B) IES retention score distribution after Ezl1, Fire1/2, or ISWI1 silencing, for IESs with an IRS > 0.1. Histogram bars with more than 10,000 IESs have been truncated for clarity. Note that the cutoff is different from Fig. 3A. (C–E) Normalized nucleosome density plots for IESs. Nucleosome density is defined as a dimensionless value (Wang et al, 2022). Representative data of nucleosome density scores between 0 and 4, which encompasses most data, are shown. (C) Histograms depicting nucleosome density in Fire1/2 silencing. Left, nucleosome density of all IESs. Middle, the IES subset sensitive to ISWI1 silencing (IRS > 0.1) with sizes larger than 200 bp. Right, the IES subset insensitive to ISWI1 silencing (IRS < 0.01), with sizes larger than 200 bp. Additional plots of different IES subsets or without IES size constraints are summarized in Appendix Fig. S8. (D) Histograms of nucleosome density from the same IES subsets as in (C), upon ISWI1 silencing. (E) Boxplots depicting nucleosome densities in the EV control, Fire1/2 and ISWI1 silencing. Data from the same IES subsets used in (C, D) are shown. The bold line denotes the median (center), the lower and upper bounds of box the 25th and 75th percentiles and the whiskers extend to the largest and smallest values no larger than 1.5× inter-quartile range (IQR). Values outside of the boundary of the whiskers are shown as outliers. The y axes have been truncated for clarity. $P_{adj}$ values were calculated by Mann–Whitney U test with Holm–Bonferroni correction (n.s. >0.01; **<0.01; ****<0.0001). The exact $P_{adj}$ values from left to right: All IESs, 2.147682e-64, 8.730297e-30, 1.936985e-03, 1.785911e-27, 1.51487e-07, 8.62239e-07; ISWI-dependent IESs, 1.342832e-47, 1.559508e-32, 6.337949e-02, 2.299329e-26, 2.522468e-10, 8.692768e-05; ISWI1-independent IESs, 1.799333e-202, 1.922333e-147, 5.723814e-01, 0.5791908, 0.2698727, 0.1806010. Number of IESs (n) from left to right: ISWI-dependent IESs. 42260, 42570, 42165, 1932, 1932, 1932; ISWI1-dependent IESs, 14055, 14125, 14049, 1347, 1348, 1348; ISWI1-independent IESs, 12964, 13102, 12922, 96, 96, 96. See also Appendix Fig. S8. Source data are available online for this figure.

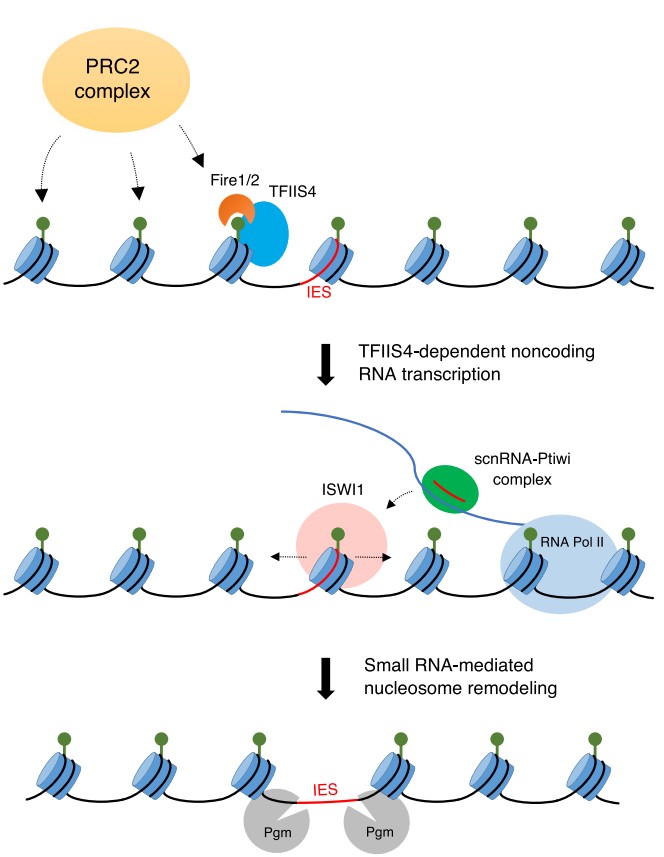

**Figure 6.  Proposed model of sRNA-mediated IES elimination.**

The PRC2 complex sets H3K9me3 and H3K27me3 in the developing new MACs, one or both of which is read by the Pc-family proteins Fire1/2. Fire1/2 acts in tight association with the transcription elongation factor TFIIS4 to generate ncRNA transcripts that are used as scaffolds for targeting by sRNAs. Targeting by sRNAs recruits the nucleosome remodeler ISWI1, resulting in nucleosome depletion on IESs. Finally, nucleosome-poor IESs can be precisely removed by the excisase Pgm.

machinery, a precision that cannot be conferred by histone modifications alone. Furthermore, the recent discovery that IESs are nucleosome-poor, a feature requiring the nucleosome remodeler ISWI1, also conflicts with this hypothesis (Singh et al, 2022). In this study, we discovered the mechanism behind PRC2-dependence on sRNA-mediated IES elimination and reconcile these seemingly contradictory findings into one unified model (Fig. 6). We demonstrated that the PRC2 complex is linked to ncRNA transcription through a tight association of the Pc-family proteins Fire1/2 and the transcription elongation factor TFIIS4. As these transcripts are believed to act as scaffolds for targeting by the sRNA machinery this can explain the requirement of PRC2 for the removal of sRNA-dependent IESs. Our findings provide evidence in support of a model mechanistically analogous to Rhi-dependent piRNA cluster transcription in *Drosophila* ovaries by utilizing a heterochromatin-dependent transcription machinery to change the mode of transcription. The tight cooperation of a histone methylation reader with a transcription elongation factor allows for heterochromatin-dependent ncRNA transcription without triggering coding transcription. This is necessary as the germline contains ~45,000 IESs scattered across the genome, which must be removed before coding transcription can take place as they often interrupt genes (Arnaiz et al, 2012).

Nucleosome profiling in the absence of Fire1/2 revealed a connection between heterochromatin-dependent ncRNA transcription and sRNA-mediated nucleosome depletion. In the absence of Fire1/2, we observed higher nucleosome densities across IESs, suggesting that open chromatin is not a pre-requisite for transcription, but a consequence of it or downstream effects. Our recent identification of the nucleosome remodeler ISWI1 and its requirement for nucleosome depletion and IES elimination provides the link between these processes (Singh et al, 2022). These findings suggest that sRNAs target IESs for elimination by dynamically changing the local chromatin architecture on or around IESs to facilitate DNA elimination. By investigating the effect of Fire1/2 depletion on the nucleosomal landscape, we were able to connect the PRC2 complex, ncRNA transcription and ISWI1-mediated nucleosome depletion into one pathway. Moreover, we found Fire1/2 to have the strongest impact on longer IESs, which may reflect a higher probability of such IESs being covered by a nucleosome and requiring nucleosome remodeling before removal.

translate well to the elimination of longer TEs in *Paramecium*, this does not seem to be the case for IESs (Wang et al, 2022). For instance, the vast majority of IESs in *Paramecium* are shorter than the size of a nucleosome and precisely excised by the excision

The PRC2 complex has dual methyltransferase activity and was shown to set both H3K9me3 and H3K27me3 during development (Frapporti et al, 2019). Despite deposition by the same methyltransferase, we identified two subfamilies of chromodomain proteins in our study: the H3K9me3 readers May1-4 and the putative H3K27me3 readers Fire1/2. We found that they are required for the removal of distinct, albeit overlapping, subsets of PRC2-dependent IESs. Unlike Fire1/2 which we found to be involved in ncRNA transcription, May1-4 is required for the removal of a subset of predominantly longer IESs. Moreover, the localization pattern of May1-4 into dynamic nuclear foci throughout new MAC development may suggest an involvement downstream of ncRNA transcription, perhaps through phase separation. Of note, the localization pattern of May1 resembles that of a well-characterized chromodomain protein in *Tetrahymena*, Pdd1 (Madireddi et al, 1996). The formation of nuclear condensates on IESs in this organism requires phosphorylation of Pdd1 as well as dimerization through its CSD (Kataoka and Mochizuki, 2015). Similarly, we found May1 immunopurified from *Paramecium* lysate to give rise to two closely spaced bands on an SDS-PAGE, which we speculate may be due to phosphorylation. Moreover, we noted a second band during the purification of recombinant May1-His roughly the size of a dimer, which was also able to bind H3K9me3 in binding assays. Hence, the similarities between May1 and Pdd1 may suggest similar chromatin dynamics during IES elimination in these ciliates.

Based on our findings, Fire1/2 and May1-4 appear to perform different functions in the IES elimination process. This surprising result argues that the roles of the PRC2 complex cannot be viewed as a single phenotype, but rather a combination of at least two distinct functions. Therefore, further investigation into these reader proteins will be beneficial to dissect the roles of the PRC2 complex as it allows for the separation of these functions. Our findings also bring forth the question of how one machinery can set two histone modifications, which likely have different downstream effects. Of note, the PRC2 complex, H3K27me3 and Fire1/2 are not only found in the new MACs but are also found in the maternal MAC in the early stages of development, unlike H3K9me3 and May1-4 that are exclusively found in the developing new MACs in the late stage (Wang et al, 2022). How can the activity of the PRC2 complex be regulated to only deposit one methylation mark in the maternal MAC? An intriguing possibility is that the presence of additional subunits controls the selective methyltransferase activity of the complex. While no such cofactor has been identified, one of the accessory subunits of PRC2 we identified in our previous study, Rnf1, was required for deposition of H3K27me3 in the maternal MAC but did not affect H3K9me3 or H3K27me3 in the developing new MACs (Wang et al, 2022). Although this likely reflects an inability of the complex to enter the maternal MAC rather than its activity per se, it opens the possibility of additional cofactors regulating different aspects of this pathway. Nonetheless, the precise mechanism and potential cofactors involved in selective methylation remains to be identified.

Our findings indicate that one or both of the "repressive" histone modifications set by the PRC2 complex fulfill an essential role as a transcriptional activator through Fire1/2 during sexual development in *Paramecium*. While we found that Fire1/2 depletion affected the deposition of both H3K9me3 and H3K27me3 in the new MACs, sequence alignments of their CDs suggested a preference for H3K27me3 but not H3K9me3. Considering that Fire and May proteins have different localization patterns and affect distinct, but overlapping, subsets of PRC2-dependent IESs, Fire and May are likely not binding to the same modification. Moreover, if they were binding the same modification and were partly redundant, the double silencing of Fire and May should result in an increased number of IESs that are retained, which we did not observe. While we cannot exclude that Fire proteins have dual binding preferences and bind to both H3K9me3 and H3K27me3, we hypothesize that the transcriptional activation function of PRC2 occurs through H3K27me3. Intriguingly, H3K27me3 was recently shown to be present on highly expressed genes during vegetative growth (i.e., binary fission) and on siRNA-targeted genes that are still being transcribed (Drews et al, 2022; Gotz et al, 2016). In light of our collective findings in this study and the previous studies, we propose that H3K27me3 is not a repressive heterochromatin mark in *Paramecium* but may rather activate transcription. In the related ciliate *Tetrahymena*, a PRC complex known as the Ezl1-complex is also required for IES elimination by deposition of H3K9me3 and H3K27me3 (Liu et al, 2007; Xu et al, 2021). The deposition of these methylation marks is required to initiate ncRNA transcription of IESs that are used as templates to generate a secondary class of sRNAs known as Late-scnRNAs (i.e., iesRNAs in *Paramecium*) (Noto et al, 2015). However, the precise mechanism controlling this transcription is unknown, and the functional distinction between H3K9me3 and H3K27me3 is unclear. If ncRNA transcription is initiated by a heterochromatin-dependent machinery as we have found in *Paramecium* and this is indeed achieved through H3K27me3, this modification may also act as a transcriptional activator in this ciliate. In favor of this hypothesis, a set of proteins known as "boundary-protecting factors" confine heterochromatin formation and ncRNA production (Suhren et al, 2017). This includes the H3K27me3 demethylase Jmj1, suggesting an involvement of H3K27me3 in this process (Chung and Yao, 2012). Our findings suggest that May and Fire proteins are not redundant but perform different functions in DNA elimination, and we propose H3K27me3 as a putative activating mark in ciliates.

## Methods

### Reagents and tools table

| Reagent/resource | Reference or source | Identifier or catalog number |
|---|---|---|
| **Experimental models** | | |
| *Paramecium tetraurelia*, strain 51 | Gift from Eric Meyer (ENS, Paris) | N/A |
| **Recombinant DNA** | | |
| Firefly1-GFP-pGEM T | This study | N/A |
| Firefly2-GFP-pGEM T | This study | N/A |
| Mayfly1-GFP-pGEM T | This study | N/A |
| TFIIS4-GFP-pGEM T | This study | N/A |
| Firefly1/2 silencing construct | This study | N/A |
| Mayfly1-4 silencing construct | This study | N/A |
| TFIIS4 silencing construct | This study | N/A |

| Reagent/resource | Reference or source | Identifier or catalog number |
|---|---|---|
| Codon-optimized May1-His for *E. coli* expression | This study; see Appendix Table S3 | N/A |
| **Antibodies** | | |
| Anti-trimethyl-Histone H3 (Lys27) | Millipore | Cat# 07-449; RRID: AB_310624 |
| Anti-trimethyl-Histone H3 (Lys9) | Millipore | Cat# 07-442; RRID: AB_310620 |
| Goat anti-rabbit Alexa Fluor 546 Secondary antibody | Thermo Fisher Scientific | Cat# A-11071; RRID: AB_2534115 |
| **Oligonucleotides and other sequence-based reagents** | | |
| Fire1 probe (for NB) | This study | N/A |
| TFIIS4 probe (for NB) | This study | N/A |
| 17S rRNA probe (for NB) | This study | N/A |
| **Chemicals, enzymes, and other reagents** | | |
| Q5 high-fidelity DNA polymerase | NEB | Cat# M0491L |
| GoTaq G2 DNA polymerase | Promega | Cat# M7848 |
| RadPrime DNA Labeling System | Invitrogen | Cat# 18428011 |
| Wheat grass powder | Pines International, Lawrence, KS | NA |
| β-sitosterol | Calbiochem, Millipore | Cat# 567152 |
| Tri Reagent | Sigma-Aldrich | Cat# T9424 |
| **Software** | | |
| ParTIES | Denby Wilkes et al, 2016 | https://github.com/oarnaiz/ParTIES |
| after_ParTIES | Swart et al, 2017 | https://github.com/gh-ecs/After_ParTIES |
| Hisat2 | Kim et al, 2019 | https://github.com/DaehwanKimLab/hisat2 |
| IES nucleosome profiling pipelines | Wang et al, 2022 | https://doi.org/10.5281/zenodo.6949086 |
| **Other** | | |
| EZ Nucleosomal DNA Prep Kit | ZYMO Research | Cat# D5220 |
| QIAGEN Plasmid Midi Kit | QIAGEN | Cat# 12143 |
| Illumina TruSeq DNA Nano Kit | Illumina | Cat# 20015965 |
| Illumina TruSeq small RNA kit | Illumina | Cat# RS-200-0012 |
| Wizard® SV Gel and PCR Clean-Up System | Promega | Cat# A9282 |
| Slide-A-Lyzer™ G2 Dialysis Cassettes | Thermo Fisher Scientific | Cat# 87723 |
| Amicon Ultra-2 Centrifugation Filter Unit | Millipore | Cat# UFC200324 |
| Ultrafree-MC Centrifugal Filter | Millipore | Cat# UFC30GV25 |

## Experimental model and strain

*Paramecium tetraurelia* strain 51 mating type seven was used in this study. Cells were grown at 27 °C in wheat grass powder (WGP) medium (Pines International, Lawrence, KS) bacterized with *Klebsiella pneumoniae* and supplemented with 0.8 mg/ml β-sitosterol. Autogamy was induced by starvation.

## GFP constructs

Genes including putative UTRs of TFIIS4 (248 bp upstream and 126 bp downstream), Fire1 (428 bp upstream and 30 bp downstream) and May1 (298 bp upstream and 361 bp downstream) were inserted into the pGEM-T vector or the pCE2 vector. TFIIS4 and May1 were N terminally tagged with GFP, and Fire1 was C terminally tagged with GFP.

## Macronuclear transformation by microinjection

Transformation of linearized GFP-tagged constructs was performed by microinjection as previously described (Beisson et al, 2010a). The successful transformation was confirmed by microscopy or dot blot as previously described (Arambasic et al, 2014).

## Cytological staging

Cells were staged by observing their nuclear morphology using 4,6-diamidino-2-phenylindole (DAPI)-staining and a minimum of 100 cells were staged per time point. The timepoints used in this study were: Early (20–40% fragmentation), Late (majority of cells with new MACs, about 6 h after 100% fragmentation) and post-autogamous (after completion of autogamy).

## Silencing by feeding

Silencing by feeding was performed as previously described using *Escherichia coli* strain HT115 (DE3) expressing double-stranded RNA to the gene of interest (Beisson et al, 2010b). All constructs were cloned into the L4440 vector between inverted T7 promoters. For co-silencings of ohnologues in the initial silencing screen, the silencing medium was mixed 1:1 according to the $OD_{600}$ prior to feeding. To increase the silencing efficiency when working with several closely related ohnologues, subsequent experiments involving Fire1/2 and May1-4 used silencing constructs generated from the concatenated open reading frames of Fire1/2, and the concatenated open reading frames of May1-4. Silencing constructs of TFIIS4 composed of fragments spanning 1–943 bp or 195–1115 bp. The empty vector (EV) of the L4440 vector was used as a negative control.

## Macronuclear DNA extraction and sequencing

Macronuclear DNA for high-throughput sequencing was extracted from 2 to 3 million post-autogamous cells, as previously described (Arnaiz et al, 2012). Cells were collected, washed twice in 10 mM Tris-HCl (pH 7.4), and resuspended in 2.5 volumes of lysis buffer 1 (0.25 M sucrose, 10 mM $MgCl_2$, 10 mM Tris-HCl (pH 6.8), 0.2% Nonidet P-40). The cell suspension was transferred to a

Potter-Elvehjem homogenizer and lysed to release the nuclei, which was assessed by DAPI staining. Following two washes with wash buffer (0.25 M sucrose, 10 mM MgCl$_2$, 10 mM Tris-HCl (pH 7.4)), the pellet was resuspended in three volumes of sucrose buffer (2.1 M sucrose, 10 mM MgCl$_2$, 10 mM Tris-HCl (pH 7.4)). A sucrose gradient was prepared by layering the samples on top of 3 ml of sucrose buffer and filling up the tube with wash buffer. Centrifugation used a Beckman Optima L-90K Ultracentrifuge at 35,000 rpm for 1 h and 4°C. After ultracentrifugation, the pellet was resuspended in 500 μl of 10 mM Tris-HCl (pH 7.4) supplemented with 10 mM MgCl$_2$, before the nuclei were lysed in 3 volumes of lysis buffer 2 (0.5 M EDTA (pH 9), 1% N-lauryl sarcosine sodium, 1% SDS, 1 mg/ml Proteinase K) and incubated at 55 °C overnight. The DNA was extracted by phenol-chloroform, dialyzed in a Slide-A-Lyser Dialysis cassette (Thermo Scientific) against 10 mM Tris-HCl (pH 8) with 1 mM EDTA (pH 8), and concentrated using an Amicon Ultra-2 Centrifugation Filter Unit (Millipore). Library preparation and sequencing was performed at the Next Generation Sequencing (NGS) Platform of the University of Bern, Switzerland. Library preparation used the Illumina TruSeq DNA PCR-Free kit, and the libraries were sequenced on a NovaSeq (paired-end 2 × 150 bp).

## Nucleosomal DNA extraction and sequencing

Nucleosomal DNA extraction was performed as previously described (Wang et al, 2022). Macronuclei from 1.5 million cells were harvested as described in the section "Macronuclear DNA extraction and sequencing" by ultracentrifugation. Following ultracentrifugation, the pellet was resuspended in 1× PBS (pH 7.4), transferred to a 1.5-ml Eppendorf tube, and washed twice with cold 1× PBS (pH 7.4). The following steps used the EZ Nucleosomal DNA Prep Kit (ZYMO Research). Nuclei were incubated with Nuclei Prep Buffer for 5 min on ice, washed twice with Atlantis Digestion buffer, and resuspended in 200 μl Atlantis Digestion Buffer. Seventeen microliters of Atlantis dsDNAse were added to the solution, and the DNA was digested for 25 min at 42 °C. The reaction was stopped by the addition of 1× MN Stop Buffer, and nucleosomal DNA was extracted using the columns supplied in the kit. Library preparation used the Illumina TruSeq Nano kit without fragmentation or size selection, and paired-end 2 × 150 bp sequencing was performed on a Novaseq. Library preparation and sequencing was performed at Fasteris (Geneva, Switzerland).

## Immunofluorescence

Around 150,000 cells were collected and washed twice in 10 mM Tris-HCl (pH 7.5). The cells were permeabilized with 1% Triton X-100 in 1× PHEM buffer (10 mM EGTA, 25 mM HEPES, 2 mM MgCl$_2$, 60 mM PIPES (pH 6.9)) for 10 min and then fixed with 2% paraformaldehyde for 10 min. After washing with 1× PBS, the cells were blocked in 3% BSA in TBSTEM buffer (10 mM EGTA, 2 mM MgCl$_2$, 0.15 M NaCl, 10 mM Tris, 1% Tween 20 (pH 7.4)) for an hour, followed by overnight incubation at 4 °C with anti-trimethyl-Histone H3 (Lys27) (07-449, Millipore) or anti-trimethyl-Histone H3 (Lys9) (07-442, Millipore) at a 1:200 dilution. Then, the cells were washed three times with 3% BSA in TBSTEM and incubated

for an hour with a goat anti-rabbit Alexa Fluor 546 Secondary antibody (A-11071, Invitrogen) at a dilution of 1:4000. Lastly, the cells were washed three times with 1× PBS, spread on glass slides and mounted with ProLong Glass Antifade Mountant (P36980, Invitrogen) containing the blue DNA stain NucBlue. Imaging was performed with an Axio Imager D2 (Zeiss) and processed with the ZEN2 software (Zeiss).

## GFP localization and imaging

Cells were collected, washed twice in 10 mM Tris-HCl (pH 7.4), and fixed in 70% EtOH. For imaging, cells were washed three times with 1× PBS (pH 7.4) and stained with DAPI. Microscopy and imaging were performed using a phase-contrast inverted microscope (Axiovert A1 or Imager.D2, Zeiss) and processed with the ZEN2 software (Zeiss).

## Survival test

Survival of post-autogamous progeny was assessed by transferring 30 post-autogamous cells to individual wells containing 0.2× WGP supplemented with 0.8 mg/ml of β-sitosterol (Merck) and bacterized with Klebsiella Pneumoniae. The cells were then monitored for 3 consecutive days and categorized into Healthy, Sick or Dead, based on the number of cell divisions.

## Total RNA extraction and sequencing

RNA was extracted from 600,000 cells following the TRIzol reagent BD protocol (Sigma-Aldrich). Library preparation and sequencing of total sRNAs was performed at Fasteris (Geneva, Switzerland). Size selection was done by polyacrylamide gel purification (17–35 nt), and subsequent library preparation used the Illumina TruSeq small RNA kit. Sequencing was performed on a NextSeq (single-end 1 × 75 bp).

## Northern blot

Total RNA (10 μg) was resolved in a 1.2% denaturing agarose gel with 2.2 M formaldehyde. The RNA was transferred onto a charged nylon membrane (Amersham Hybond-XL, GE Healthcare Life Sciences) by capillary blotting in 20× saline-sodium citrate (SSC) buffer, followed by 2× UV crosslinking at 120,000 μJ/cm². The membrane was washed once in 2× SSC buffer for 10 min and prehybridized in Church buffer (1% BSA, 0.5 M NaPO$_4$ (pH 7.2), 7% SDS, 1 mM EDTA) for 2 h at 60 °C. Hybridization was performed in Church buffer containing radioactive probes specific to TFIIS4, Fire1 or 17S rRNA (see below), and incubated overnight at 60 °C. After hybridization, the membrane was washed twice with 2× SSC buffer containing 0.1% SDS and exposed on a phosphor screen (Amersham). Visualization was performed on a Typhoon FLA 7000 (GE Healthcare).

Probes: α-P$^{32}$ dATP-labeled TFIIS4 (558–943 bp) and Fire1 (279–491 bp) probes were labeled using the RadPrime DNA Labeling System (Invitrogen). A probe complementary to the 17S rRNA (ACC CGT GAC TGC CAT GGT AGT CCA ATA CA) was 5' end-labeled with γ-P$^{32}$ dATP using T4 Polynucleotide Kinase (Thermo Scientific) and used as a loading control.

## Protein expression and purification of May1-His

Codon-optimized May1-His for expression in *E. coli* was synthesized and cloned into the pET-32a(+) expression vector by Genscript (Appendix Table S3). Protein expression and purification was performed as previously described with minor modifications (Shanle et al, 2017). The construct was transformed into BL21 (DE3), and the bacteria inoculated in LB supplemented with ampicillin. After incubating for 16 h at 37 °C, the culture was diluted 1:100 in LB and incubated until it reached an $OD_{600}$ between 0.4 and 0.6. After chilling the culture at 4 °C for 30 min, 0.4 mM IPTG was added to induce the expression of the protein, followed by a 20-h incubation at 16 °C. Bacteria were harvested by centrifugation, washed in 0.85% NaCl, flash-frozen in liquid nitrogen and stored at 80 °C. For protein purification, bacterial pellets were resuspended in lysis buffer (50 mM Tris pH 8, 250 mM NaCl, 4 mM DTT, 0.1 mM phenylmethane sulfonyl fluoride (PMSF), 0.5 mg/ml chicken egg lysozyme, 0.2% Triton X-100, 10% glycerol, 1× Protease inhibitor complete tablet (Roche)) and incubated on ice for 45 min, before lysing by sonication at 4 °C. The lysate was clarified by centrifugation for 45 min at 15,000× *g* and 4 °C. Ni Sepharose High-Performance beads (Cytiva) were pre-equilibrated with binding buffer (50 mM Tris-HCl pH 8, 250 mM NaCl, 4 mM DTT, 10% glycerol) and incubated with the cleared lysate for 1 h at 4 °C on a rotor. Then, the beads were washed with 10 bed volumes of binding buffer and eluted in elution buffer containing increasing amounts of imidazole (50 mM Tris-HCl pH 8, 250 mM NaCl, 4 mM DTT, 10% glycerol, imidazole (50, 100, 150, or 200 mM)). Coomassie staining was used to estimate the protein concentration and purity of the eluates, which were then pooled. The eluate was dialyzed against 1.3 L of binding buffer at 4 °C, which was exchanged after 2 h and left overnight. Proteins were concentrated by centrifugation using PierceTM Protein Concentrators PES (Thermo Scientific). Lastly, the concentration and purity were assessed by Bradford assay (Bio-Rad) and Coomassie staining. The proteins were stored in 50% glycerol at −20 °C.

## In solution peptide pull-down assays using purified proteins

Peptide pull-down assays were performed as previously described (Shanle et al, 2017). Fifty picomol of His-tagged May1 was incubated with 500 pmol biotinylated histone peptides in 200 μl of Peptide binding buffer (50 mM Tris-HCl pH 8, 300 mM NaCl, 0.1% NP-40) for 1 h at 4 °C on a rotor. In total, 20 μl of streptavidin-coated magnetic beads (Pierce) were pre-equilibrated by washing three times for 5 min in Peptide binding buffer. After the incubation, the pre-equilibrated beads were added to the protein–peptide mixture, followed by an incubation for 1 h at 4 °C on a rotor. The beads were washed thrice with Peptide binding buffer and bound complexes were eluted by the addition of 1× SDS loading dye and boiling for 5 min at 95 °C. Bound proteins were resolved by SDS-PAGE and analyzed by western blots. Histone peptides used in this study are listed in Appendix Table S4.

## Identification of *P. tetraurelia* chromodomain proteins

Chromodomain (CD) sequences of the thirteen *T. thermophila* chromodomain proteins reported in a previous study were used to identify *P. tetraurelia* chromodomain proteins by BLAST (Appendix Table S1) (Camacho et al, 2009; Wiley et al, 2018). In brief, the CD sequences were used as a query to perform a blast search using blastp through the *Paramecium* Database (Arnaiz et al, 2020). Following this search, each candidate was manually examined to ensure the inclusion of all ohnologues (paralogs from whole-genome duplication events), since not all were identified in the initial search.

## Phylogenetic analyses of *P. tetraurelia* chromodomain proteins

Multiple alignments of full-length proteins were computed by MAFFT v7.520 with "einsi" mode (Katoh and Standley, 2013). We kept the entire alignments for downstream analyses (removal of regions with large insertions and/or deletions was not performed), as the negative impact of such filtering has been demonstrated (Tan et al, 2015). Maximum likelihood (ML) phylogenetic trees were constructed by RAxML v. 8.2.9 with the substitution model 'PROTGAMMAIAUTO' and 100 bootstrap replicates (Stamatakis, 2014). Trees were visualized by Figtree ver. 1.4.4.

For analyses of extracted CD sequences, the CD regions were identified by InterProScan (https://www.ebi.ac.uk/interpro/search/sequence/), and the annotated sequences (InterPro accession: IPR000953) were extracted for subsequent analyses (Paysan-Lafosse et al, 2023). Multiple alignments and ML trees were computed by the same method as described above. One chromodomain protein (Mayfly1) was found to carry two putative CDs; however, further analysis identified the second as a chromo shadow domain (CSD) and not a CD. Our initial approach including the full set of domains showed an extremely long branch of this CSD, and to avoid long branch attraction, the CSD was removed from the datasets for the final analysis.

## Alignment of chromodomains

To generate an alignment with *Drosophila* HP1 and Pc CDs, amino acid sequences of the CDs of Fire1-2, May1-4, *Drosophila* HP1 and Pc were manually cropped and aligned using MUSCLE (v3.8) through EMBL-EBI (https://www.ebi.ac.uk/Tools/msa/muscle/) using default parameters (Edgar, 2004; Madeira et al, 2022).

## Reference genomes

The following reference genomes were retrieved from the *Paramecium* Database (Arnaiz et al, 2020) and used for IES retention score calculations and sRNA mapping: MAC (mac_51), MAC + IES (mac_51_with_ies), TE (ptetraurelia_TE_consensus_v1.0) and OES (ptetraurelia_mic2).

## Small RNA mapping

Small RNA reads were mapped to the reference genomes by iterative read alignment with Hisat2 and read subtraction (Kim et al, 2019). Histograms were generated by normalizing mapped sRNA reads by the total number of reads and plotting each size separately.

## IES retention scores and correlation plots

IES retention scores (IRS) were calculated using ParTIES, and correlation plots were generated using After_ParTIES (Denby

Wilkes et al, 2016; Swart et al, 2017). An IRS table containing IRS of all IESs for various silencings used in this study can be found in the supplement (Dataset EV2).

## Nucleosome density analysis

Relative nucleosome densities on IESs were calculated as previously reported (Wang et al, 2022). Briefly, adapters and low-quality reads were filtered out using bbduk in the BBTools package (v38.96) with flags "ktrim=r k = 20 qtrim=rl trimq=15 minlen=50 ftr=90 tpe tbo". Hisat2 (v2.1.0) was used to map the libraries to the MAC + IES reference genome with flags "--min-intronlen 24 --max-intronlen 20000 --no-mixed" (Kim et al, 2019). Concordant, primary-aligned pairs were selected using samtools v1.13 (options "-f2 -F256") (Danecek et al, 2021; Kim et al, 2019). MNase-treated samples were further filtered to the fragment size of mononucleosomes (125–175 bp). Libraries were subsequently downsampled so that the number of reads among MAC DNA libraries, and those among nucleosomal DNA libraries, were comparable. The nucleosome occupancy over IESs was calculated by Bedtools v2.30.0 and HTSeq v1.99.2 (Anders et al, 2015; Quinlan and Hall, 2010). The final HTSeq output is available in the supplement (Dataset EV3). Relative nucleosome density across IESs was defined as normalized MNase-read divided by normalized input-read. Detailed formulas and scripts are described and available in (Wang et al, 2022). Our previously published control libraries (EV/PGM-KD MAC and EV/PGM-KD nucleosome) and ISWI1-silencing libraries (ISWI/PGM-KD MAC and ISWI/PGM-KD nucleosome) were re-analyzed together with the FIRE/PGM-KD libraries obtained in this study (Singh et al, 2022; Wang et al, 2022). Statistical significance of the differences in nucleosome density scores among samples (Fig. 5E) were calculated by the Mann–Whitney $U$ test with Holm–Bonferroni correction.

## Data availability

Next-generation sequencing data generated by this study have been deposited to NCBI under the BioProject ID PRJNA899513 and are publicly available as of the date of publication.

The source data of this paper are collected in the following database record: biostudies:S-SCDT-10_1038-S44319-024-00332-1.

## Peer review information

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

## Acknowledgements

The authors thank Dr. Nassikhat Stahlberger for technical support and the entire Nowacki lab for discussions. Computations were partially performed on the NIG supercomputer at ROIS National Institute of Genetics, Japan. This work was supported by European Research Council Grants (ERC) 260358 "EPIGENOME" and 681178 "G-EDIT", Swiss National Science Foundation Grants 31003A_146257 and 31003A_166407, and grants from the National Center of Competence in Research (NCCR) RNA and Disease to MN. This work was supported by the World Premier International Research Center Initiative (WPI), MEXT, Japan.

## Author contributions

**Therese Solberg**: Conceptualization; Formal analysis; Supervision; Investigation; Visualization; Writing—original draft; Writing—review and editing. **Chundi Wang**: Investigation; Visualization. **Ryuma Matsubara**: Data curation; Formal analysis; Investigation; Visualization; Writing—review and editing. **Zhiwei Wen**: Investigation. **Mariusz Nowacki**: Conceptualization; Supervision; Funding acquisition; Project administration; Writing—review and editing.

Source data underlying figure panels in this paper may have individual authorship assigned. Where available, figure panel/source data authorship is listed in the following database record: biostudies:S-SCDT-10_1038-S44319-024-00332-1.

## Disclosure and competing interests statement

The authors declare no competing interests.

# Expanded View Figures

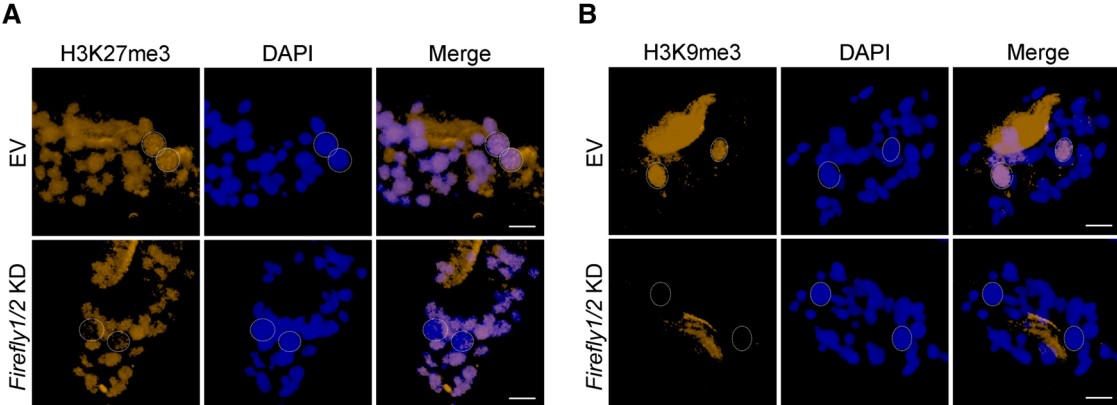

**Figure EV1.  Fire1/2-KD affects the deposition of H3K27me3 and H3K9me3 in new MACs.**

(**A**) Immunofluorescence of H3K27me3 at the late stage of development in EV and Fire1/2-silenced cells. (**B**) Immunofluorescence of H3K9me3 at the late stage of development in EV and Fire1/2-silenced cells. Dotted circles denote new MACs. Scale bars: 10 μm.

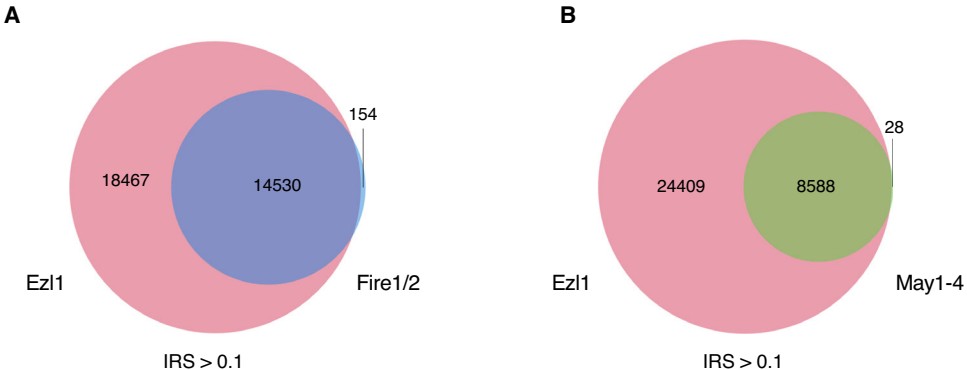

**Figure EV2. Shared IES retention between Ezl1, Fire1/2 and May1-4 silencing.**

(A) Venn diagram depicting shared IES retention between Ezl1 and Fire1/2 silencing (IRS > 0.1). (B) Venn diagram depicting shared IES retention between Ezl1 and May1-4 silencing (IRS > 0.1).

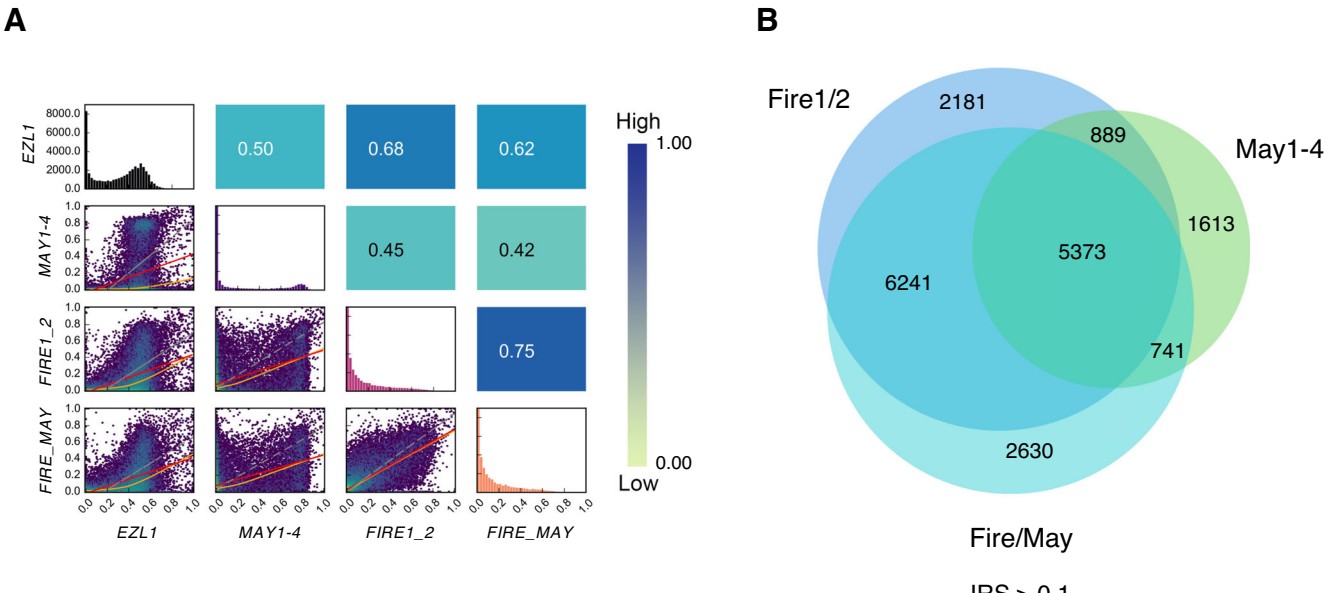

**Figure EV3. Correlation and shared IES retention between May1-4, Fire1/2 and Fire/May-KD.**

(A) Correlation plots calculated by hexagonal binning of IES retention scores generated using After_ParTIES (Swart et al, 2017) and the IES retention scores provided in Dataset EV2. Pearson's correlation coefficients are given above each subgraph. Red lines are for ordinary least-squares (OLS) regression, orange lines for LOWESS, and gray lines for orthogonal distance regression (ODR). From light green to dark blue, the correlation is stronger. (B) Venn diagram depicting shared IES retention between Fire1/2, May1-4 and Fire1/2/May1-4 ("Fire/May") silencing (IRS > 0.1).

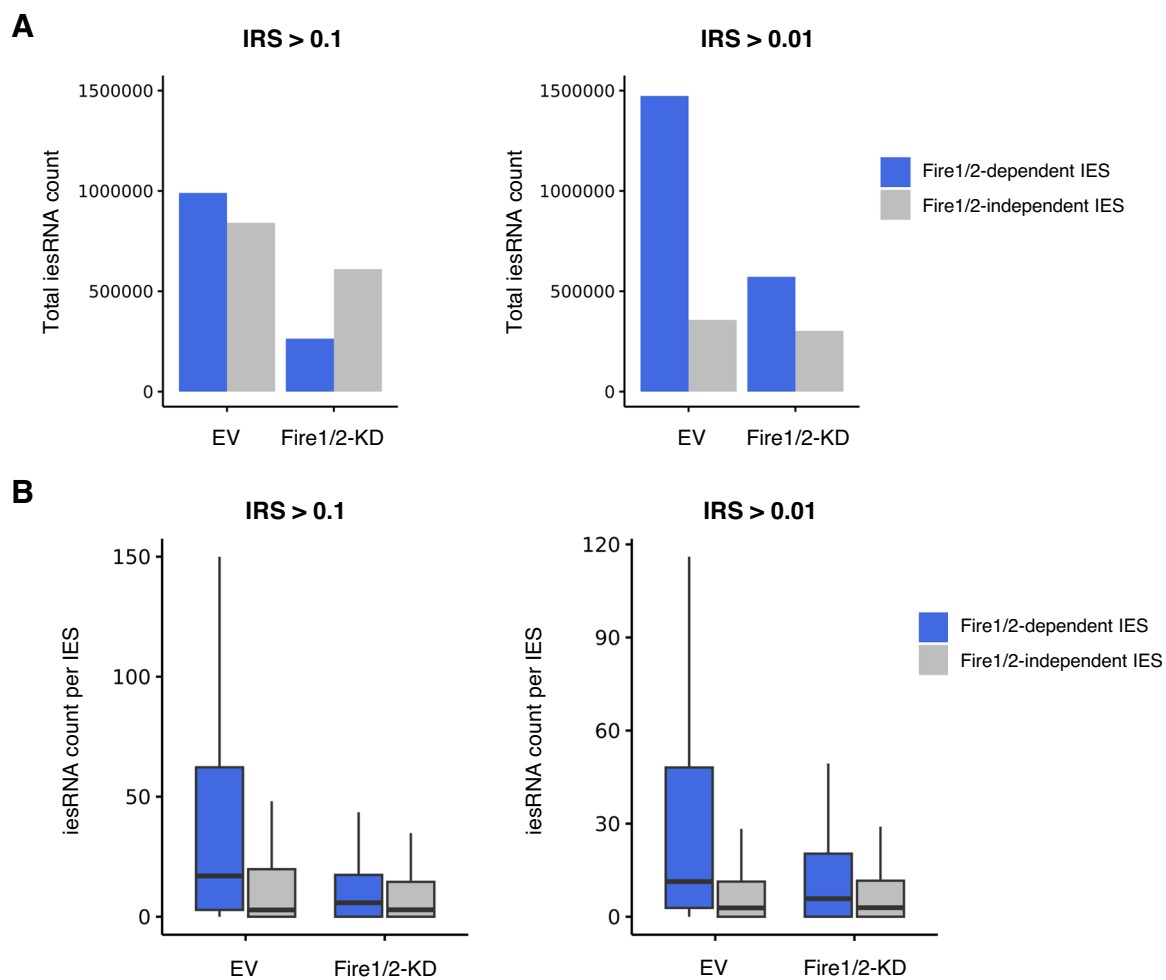

**Figure EV4. iesRNAs produced from Fire1/2-dependent and independent IESs.**

(A) Total iesRNA counts corresponding to Fire1/2-dependent or independent IESs, with a retention score cutoff of 0.1 (left) or 0.01 (right) to denote Fire1/2-dependent IESs. (B) iesRNA count per IES for Fire1/2-dependent or independent IESs, with a retention score cutoff of 0.1 (left) or 0.01 (right) to denote Fire1/2-dependent IESs. The bold line denotes the median, the lower and upper hinges the 25th and 75th percentiles and the whiskers extend to the largest and smallest values no larger than 1.5 x inter-quartile range (IQR). Outliers were omitted for better visualization. Number of IESs (n) from left to right: IRS > 0.1, 14684, 30243, 14684, 30243; IRS > 0.01, 26840, 18087, 26840, 18087. For all plots, 21 to 24 and 26 to 30 nt sRNA with perfect IES-matching sequences were selected as iesRNAs, and the reads were normalized to the 23 nt siRNAs mapped to the backbone of the L4440 vector.

