## [Peer Review File · EMBO Reports]

Heterochromatin-dependent transcription links the PRC2 complex to small RNA-mediated DNA elimination

Therese Solberg, Chundi Wang, Ryuma Matsubara, Zhiwei Wen, and Mariusz Nowacki

Corresponding author(s): Mariusz Nowacki (mariusz.nowacki@unibe.ch) , Therese Solberg (therese.solberg@keio.jp)

Review Timeline:

Submission Date:	13th Jun 24
Editorial Decision:	17th Jun 24
Revision Received:	30th Sep 24
Editorial Decision:	4th Nov 24
Revision Received:	12th Nov 24
Accepted:	15th Nov 24

Editor: Esther Schnapp

Transaction Report:

Dear Mariusz,

Thank you for the transfer of your manuscript and referee reports to EMBO reports. We have also received comments on your proposed revision plan from the referees, which are pasted below.

As you will see, both referees 1 and 3 agree with your revision plan and that the study is a good fit for EMBO reports. I would therefore like to invite you to revise the ms along the lines suggested by you and the referees.

Please note that the referee concerns must be fully addressed and their suggestions taken on board. Please address all referee concerns in a complete point-by-point response. Acceptance of the manuscript will depend on a positive outcome of a second round of review. It is EMBO reports policy to allow a single round of major revision only and acceptance or rejection of the manuscript will therefore depend on the completeness of your responses included in the next, final version of the manuscript.

We realize that it is difficult to revise to a specific deadline. In the interest of protecting the conceptual advance provided by the work, we recommend a revision within 3 months (17th Sep 2024). Please discuss the revision progress ahead of this time with the editor if you require more time to complete the revisions.

- 1) A data availability section providing access to data deposited in public databases is missing. If you have not deposited any data, please add a sentence to the data availability section that explains that.
- 2) Your manuscript contains statistics and error bars based on $n=2$. Please use scatter blots in these cases. No statistics should be calculated if $n=2$.

5) a complete author checklist, which you can download from our author guidelines <https://www.embopress.org/page/journal/14693178/authorguide>. Please insert information in the checklist that is also reflected in the manuscript. The completed author checklist will also be part of the RPF.

6) Please note that all corresponding authors are required to supply an ORCID ID for their name upon submission of a revised manuscript (<https://orcid.org/>). Please find instructions on how to link your ORCID ID to your account in our manuscript tracking system in our Author guidelines <https://www.embopress.org/page/journal/14693178/authorguide#authorshipguidelines>

7) Before submitting your revision, primary datasets produced in this study need to be deposited in an appropriate public database (see <https://www.embopress.org/page/journal/14693178/authorguide#datadeposition>). Please remember to provide a reviewer password if the datasets are not yet public. The accession numbers and database should be listed in a formal "Data

Availability" section placed after Materials & Method (see also <https://www.embopress.org/page/journal/14693178/authorguide#datadeposition>). Please note that the Data Availability Section is restricted to new primary data that are part of this study. * Note - All links should resolve to a page where the data can be accessed. *

12) All Materials and Methods need to be described in the main text. We would encourage you to use 'Structured Methods', our new Methods format. According to this format, the Methods section should include a Reagents and Tools Table (listing key reagents, experimental models, software and relevant equipment and including their sources and relevant identifiers) followed by a Methods and Protocols section in which we encourage the authors to describe their methods using a step-by-step protocol format with bullet points, to facilitate the adoption of the methodologies across labs. More information on how to adhere to this format as well as downloadable templates (.docx) for the Reagents and Tools Table can be found in our author guidelines: < <https://www.embopress.org/page/journal/14693178/authorguide#manuscriptpreparation>>.

An example of a Method paper with Structured Methods can be found here: <https://www.embopress.org/doi/full/10.1038/s44320-024-00037-6#sec-4>

You are able to opt out of this by letting the editorial office know (emboreports@embo.org). If you do opt out, the Review Process File link will point to the following statement: "No Review Process File is available with this article, as the authors have

chosen not to make the review process public in this case."

I look forward to seeing a revised form of your manuscript when it is ready.

Comments on your proposed revision plan:

Referee 1:

Me and both additional referees asked for additional experiments to support the conclusions: ChIP Seq and Gro Seq of new Macs. The authors explain logically, why these techniques cannot applied to the current problem, in both cases due to high degree of contamination of the old Mac, due to the impossibility to synchronize cultures. I agree that this is Paramecium specific issue which cannot be solved right now.

The suggested IF measurement of H3K27me3 is used in many other studies and will improve the paper.

Going through the comments and plans of the authors, I think that the entire story is worth publishing and your Journal fits to this quite well.

The Paramecium system does not allow for a direct transfer of approaches from other systems, but its clear that the authors need to be much more careful to formulate conclusions out of the data situation: the latter will surely be improved according to their plans. I think also referee 3 does not insist on additional data, asking either for additional experiments OR to modify the models and conclusions.

In general, the paper is now written from the point of view on the role of H3K27me3 - a mark which is under debate in Paramecium and referee 2 is right, that for this more data, maybe in vitro data on enzymes, is necessary to generate biochemical evidence, either by binding assays, or run on transcription - but the authors claim that this is difficult. To change point of view avoiding overstatements on the histone mark, but to focus on Fire and May function during development and IES excision, with the additional aspect offered to analyze not only scnRNAs but also IES RNAs could be a solution to answer more specific questions and to provide suitable, data supported models for the Paramecium system. I do not think that the paper will then still be of a broad interest without going too much into ciliate genetics, this is why I asked for more comparisons to other species.

I would recommend to proceed with the experiments and to rewrite the story. I feel that this will results in a interesting story which will expand the chromatin knowledge.

Referee 2:

I agree with the revision plan they propose, and I understand well that ChIP-seq is quite difficult in Paramecium. The unique reports in Paramecium from the authors and also Tetrahymena by others are important and valuable to many researchers in understanding the diverse functions of small RNAs.

However, I am not sure if a paper with conclusions that have to be toned down due to lack of experimental proof is appropriate for publication in EMBO Reports.

Rather, I think it would be more suited for publication in Life Science Alliance, or other more specific journal.

Referee 3:

After reading authors' replies to the reviewers' comments and the proposed plan, I think it is a great way to improve the manuscript. Adding immunofluorescence data and analyses of IES retention in the double-knockout mutants will both be a significant addition to the work. Authors also clearly described their failed attempts to purify recombinantly expressed proteins and should very briefly mention that in the revised version. Of note, I do not support reviewer#2's view that this study focuses on questions in "the specific area of genome rearrangement in Paramecium". This work addresses the basic biology of regulation of gene expression and how different organisms solve such problems.

Referee #1:

Solberg and co-workers study DNA elimination processes in the ciliate *Paramecium*, which together with other ciliate species provides an excellent model to study this particular genome editing mechanism, which is controlled by small RNAs derived from selective hybridisation with parental RNA. This work fills a long-standing logical gap in previous models, as histone modifications, in particular H3K27me3 and H3K9me3, have been described to be associated with developmental chromatin, but the previous hypothesis that these might be directly associated with elimination is at odds with the existence of quite short eliminated sequences of a length that cannot be associated with nucleosomes. Further studies of nucleosome occupancy have revealed quite open chromatin and low nucleosome density around eliminated sequences, further challenging the histone marking hypothesis. This paper presents histone modifications, in particular H3K27me3, in the context of ncRNA transcription, arguing against a general repressive function of this mark. These results shed massive new light on the field, not necessarily on the ciliate field, but on the general community focused on chromatin biology. The work is nicely designed with suitable controls.

The paper is well written and easy to follow. On first reading I felt that too much of the results chapter was in the style of an introduction or review, but in the end I think this makes it easier for the reader to follow. I have few comments and suggestions.

Thank you for the kind words and appreciating our work!

Major comments or suggestions:

- I think the final model in Fig. 6 is supported by the data, but I am confused about the role of H3K27me3. For transposable elements, the model of Miro-Pina et al 2022 is more like the co-transcriptional silencing model, where Piwi or Ago binds to nascent transcripts and recruits the histone modifiers. Here, in contrast, K27me3 is positioned independently of Piwi and triggers nascent transcripts that are substrates for Piwi-loaded sRNAs. Is this a difference between transposons and IESs? Would the PRC2 complex act specifically on the entire chromatin here, or do the authors think that this can create a feed-forward loop during the first rounds of endoreplication?

As the reviewer suggested, we also believe that the role of the PRC2 complex on DNA elimination differ between TEs and IESs prior to their removal. For TEs, it induces heterochromatin formation and repression, whereas for IESs, it promotes transcription and nucleosome depletion. Since these are very different effects on chromatin, we agree that this is important to point out and have included this distinction in the first paragraph of the discussion section.

As for where this mechanism takes place, we can only speculate at this point. Considering that efficient targeting of IESs scattered across the genome by sRNAs would require timely transcription from the entire new MAC genome, initiation from one particular location sounds unlikely, and we therefore rather favor a model in which this mark is set on many locations across the genome. One intriguing possibility is if this mark is set on or near IESs or other eliminated sequences, which are

then removed throughout development, effectively serving as a programmed shutoff once IES elimination is complete. However, these are so far only speculations and we do not have experimental evidence to favor one model over another. Therefore, further work is required to uncover the details of how the PRC2 complex and these methylation marks influence the chromatin.

- If I remember correctly, not all IESs concatamerise, larger ones also circularise. Given that May and Fire differ in the size of the retained IESs, do the remaining iesRNAs reflect those from either long or short IESs or other circularised or circularised and concatamerised IESs? Could this shed light on whether the transcriptional machinery of circularised IESs is also affected in Fire KD?

This is an excellent question that has also crossed our minds: are Fire-TFIS4 involved in transcription of IES concatemers? As the reviewer suggested, we have analysed the contribution of Fire1/2-dependent and independent IESs to the iesRNA pool in both the EV and Fire1/2-silenced samples. This analysis revealed that Fire1/2-dependent IESs contribute to a larger extent to the total iesRNA pool than independent IESs, likely due to the size differences. Moreover, we found that there is a much greater effect on Fire1/2-dependent than independent iesRNAs between the EV and Fire1/2-silencing, which may suggest that Fire1/2 are not involved in the transcription of IES concatemers. However, if we chose a higher IRS cutoff there was still a slight reduction of Fire1/2-“independent” iesRNAs in the silencing, which was not the case for a lower cutoff. Therefore, we decided to include two IRS size cutoffs in the figure (Figure EV4) and manuscript. Based on these results, we believe that the reduction of iesRNAs is because of the block in IES excision and that Fire1/2 are not involved in the transcription of IES concatemers.

- The pull down assay of modified histone needs further clarification. Please provide the peptide sequences and note if they are *Paramecium* sequences, just to exclude that an amino acid substitution hinders H3K27 to be detected. A reference to Figure 2H is missing from the text, at least as far as I have found.

The information about the peptide sequences used in the assay can be found in Appendix Table S4, and we have included a sentence about that they are not *Paramecium* sequences in the table legend. We have also included the missing reference to Figure 2H (Thanks!).

-Does Fire KD affect the H3K27me3 abundance? In association with the comment before and in the highlight that the KD of the reader could affect the abundance of the histone mark (readers can associate to writers) it would be good to know if the H3K27me3 level is altered in the KD.

We thank the reviewer for this suggestion. As requested, we performed immunofluorescence of H3K27me3 in control (EV) and Fire1/2-KD to assess whether or not Fire1/2 affects its abundance. To be more comprehensive and because we were unable to prove that Fire1/2 binds H3K27me3, we additionally investigated H3K9me3 in the same conditions. Interestingly, the signal of both modifications disappeared from the new MACs when Fire1/2 was silenced, which may either suggest that Fire1/2 binds to H3K27me3 but is required for the deposition of H3K9me3, or that Fire1/2 binds to both H3K9me3 and H3K27me3 (Figure EV1). We favor the first hypothesis, since Fire1/2 contain a Pc-like CD and it is likely that disrupting ncRNA transcription also affects downstream processes,

which may include the activity of May1-4. However, it is also possible that Fire1/2 have dual binding preferences for both H3K9me3 and H3K27me3. We have added these results to the manuscript and discussed the findings in the discussion section.

Minor comments:

- Page 5: about 90%: give exact number.

Thank you - we have changed it to the exact number (91.2%).

- In the introduction I miss the comparison to both co-transcriptional silencing by PolIII in yeast (and maybe transcription of heterochromatin by PolIV and V in plants). It is not new that special transcription machinery is able to transcribe heterochromatin or even "repressive" marks, although the situation here is much more intriguing. It should also be noted that increased H3K27me3 levels have previously been observed in *Paramecium* for siRNA-targeted genes that are still being transcribed (Götz et al. 2016): this shifts the focus from transcription in an on/off state to the quality of the RNA produced, and in this paper we consider ncRNA rather than mRNA.

Thank you for the comment. We agree that these are important to mention, and have included references to CTGS in *S. pombe* and the RdDM pathway *A. thaliana* to the revised introduction. The findings from Götz et al regarding H3K27me3 on siRNA-targeted genes have been added to the discussion section together with Drews et al 2022, where we discuss about the possibility of H3K27me3 acting as a transcriptional activator.

- Page 6: Mutated aromatic residues: better divergent?

Thank you - we have changed the term from mutated to divergent.

- Is heterochromatin protein 1 usually not HP1 instead of Hp1?

Indeed. We have changed the term from Hp1 to HP1 throughout the manuscript and figures accordingly. Thanks!

- Page 10: Tight association suggests physical interaction. Functional linkage seems to be more precise. In this context, the most precise way to demonstrate functional linkage would be to show altered transcription via Gro-Seq (e.g. Schoeberl et al. 2012 in *Tetrahymena*) in Fire KD lines.

While we would love to use Gro-seq to answer this question, this method has not been performed in *Paramecium*, and it would be technically very challenging if not impossible at this point due to the lack of method for isolating new MACs at this stage of development, in which the maternal MAC fragments are still transcriptionally active and have roughly the same size as the new MACs. We ask for the reviewers understanding in this regard.

- Fig.4F is of poor quality and it is questionable what it can contribute...

This picture is from a urea-PAGE of radioactively labeled small RNAs after Fire1/2-KD to visualize that the lack of iesRNAs is also visible without sequencing, but the corresponding sequencing results are shown in Figure 4G and we have therefore removed it.

- Page 11- (line numbers would have been great!) "... Indicate that scanning occurs..." please be clearer: why does it indicate that scanning occurs, or does it indicate that scn RNA normally accumulates?

This result indicates that scanning (of scnRNAs) occurs because of the enrichment of IES-matching scnRNAs relative to MAC-matching scnRNAs. scnRNAs are believed to be generated from the entire MIC genome, and during early stages of development there are more MAC-matching than IES-matching scnRNAs (Figure R1A). By the late stage of development, the scnRNAs are predominantly matching to eliminated sequences (IESs and OESs; Figure R1B), as a consequence of the selective degradation of MAC-matching scnRNAs. As a comparison, please see Figure R1D below for small RNA plots from Wang et al, 2022 (PMID: 36001962), showing the size distribution and targets of small RNAs extracted from the late-stage of development when Ezl1 is silenced and scanning does not occur. Since Ezl1-KD leads to scanning defects and Fire1/2 are also present in the maternal MAC (albeit faint), this was important to investigate to exclude an effect on scanning. We have explained this better in the manuscript.

(Also: we have included line numbers in the revision so it is easier to navigate)

Figure R1. Ezl1-KD impairs scanning (see B (EV) vs D (Ezl1-KD)).

From Wang et al., Cell Reports, 2022 (PMID: 36001962).

Referee #2:

- general summary and opinion about the principle significance of the study, its questions and findings
In Paramecium and Tetrahymena, the genome sequence is different between MIC and MAC by the action of small RNA-mediated genome rearrangement. In this process, sequences such as TE and IES are removed from the genome, and little transcription occurs in the MIC, while transcription occurs only in the MAC where rearrangement has been completed. Such genome rearrangement regulates the life of these organisms. Although many studies have been conducted on this genome rearrangement, the molecular mechanism has not yet been fully elucidated. In this paper, the authors reported a detailed description of the molecular mechanism of genome rearrangement in Paramecium. They initially have three main questions to answer. They had previously reported on the contribution of H3K27me3 to DNA rearrangement and analyzed whether specific chromatin readers are involved in this process. They sought to test the model that H3K27me3 may be involved in the promotion of transcription. Their findings from these analyses are as follows: they found specific chromodomain proteins are involved in genome rearrangement. They found some correlation of phenotypes between CD mutant and some previously known mutants.

- specific major concerns essential to be addressed to support the conclusions

The models and challenges they set are widely shared in the piRNA and heterochromatin fields as important issues, some of which have already been clarified in flies in great detail. I understand that their goal is still recognized as a problem to be solved in the specific area of genome rearrangement in Paramecium. On the other hand, the data presented in this paper are not of sufficient quality to answer the questions they set forth, and many parts of the paper overstate the data. In other words, I think their conclusions are not supported by the data. The most important parts are as follows.

- ChIP-seq was not performed to show the genomic localization of the chromatin readers, so we do not know in which region of the genome the molecular mechanism they intend is occurring
- The small RNA-seq analysis does not mention about the content of the sequences.
- Overall, the presentation of data is not reader friendly. It is not clear what kind of experiments were conducted even after reading the legend. For example, the scale of the vertical axis of the graph is unclear, and the font is too small, making it extremely difficult to read.

In sum, I think the presented data do not support their claims.

We thank the reviewer for taking the time to thoroughly review our work and the effort to improve our manuscript. In addition to including findings from several new experiments and analyses suggested by the reviewer, we have carefully revised our manuscript to tone down any overstatements and to better reflect our experimental evidence, as well as included evidence from other studies where needed.

Regarding the three important points above:

1. We agree that ChIP-seq would be a better way to show the genomic localization of the chromatin readers; however, there are multiple issues with performing ChIP seq in this organism. The first is technical: successful ChIP has not been performed for proteins in the new MACs of ciliates, and one big hurdle is that we cannot isolate the new MACs from the fragments of the maternal MAC as they are roughly the same size. Secondly, the issues of the biological samples, which differ greatly from flies. We are unable to synchronize the cells, meaning that there is always heterogeneity in the samples, which is a problem for proteins that are only present in a very narrow time window. On top of that, since the way that IESs and TEs are controlled is not through repression, but a physical elimination of the sequences, these sequences are gradually removed while the genome is amplified to 800n. This also introduces heterogeneity because of differences in the stages of the cells, and if a protein binds to IES sequences, it will be underrepresented as there are far fewer reads even in the input, which is an even bigger issue if it binds to both IES and non-IES sequences - we may never detect the IES sequences since they are much less abundant. Third, the data analysis itself is a challenge, because of the issues stated above, such as the difference in ploidy between the MIC (2n), MAC (800n) and new MACs (something between those), and that the IESs are underrepresented in the samples. This means we cannot be sure of the preference of non-IES to IES sequences.

The above points are general issues with ChIP-seq of proteins in the new MACs, but with Fire1/2 and TFIS4 there are two additional big issues. Previous evidence suggests that the ncRNAs are generated from the entire genome, not from particular loci as for the piRNA pathway in flies, which may also complicate interpretations of the ChIP seq with Fire and TFIS4, even if all the other issues could be resolved. And lastly, when attempting to immunoprecipitate Fire1 from *Paramecium* lysate (we indeed tried), the protein pelleted and was insoluble, likely due to that Fire is highly disordered (around 70% disordered). Highly disordered proteins are notoriously difficult to IP...

We hope this clarifies why the experiment suggested is not feasible, and we ask for the reviewers understanding in this regard. As the reviewer suggested in the minor concerns below in the case that the experiment was not feasible, we have toned down the statements accordingly.

2. The small RNA-seq analysis does include the content of the sequences by color-coding, and we have written this more clearly in the manuscript and the legend to avoid confusion.

3. We agree that the data presentation, in particular that of Figure 1, was lacking in the previous version of the manuscript and we have therefore improved this in the revised version, including adjusting the text to appropriate font sizes, including more details in the legends, and adding axis titles.

- minor concerns that should be addressed

•In the Introduction, you say "Fire and May depletions affect IES elimination in distinct ways", but at what level are you saying DISTINCT way? We need to isolate whether it is a genetically independent pathway or a redundant pathway. Specifically, we need to verify if the phenotype is the same as Fire KD by conducting experiments with following double mutants, "Fire KD and May KD", "Fire KD and TFIIIS4 KD", and "Fire KD and ISWI1 KD".

We agree that this is important to tease apart and we performed the experiments accordingly. The following data have been added to the manuscript and figures:

- "Fire KD and TFIIIS4 KD" correlated very strongly with the single-KDs and Dcl2/3/5-KD (Figure 4C). This was to be expected based on their mutually-dependent phenotypes.
- "Fire KD and May KD" correlated strongly with the Fire1/2 single-KD, but not with the May single-KD (because this is a smaller subset), and the retained IESs encompassed most of the IESs of the single-KDs (Figure EV3).

For "Fire KD and ISWI1 KD", we faced several issues. This silencing was weaker than the other two for unknown reasons (Figure R2), and we had an issue with DNA degradation in these samples specifically. Despite several attempts at this silencing, we were unable to improve it and therefore resorted to deep-sequence MAC genomic DNA displaying only a weak IES retention phenotype (Rep 1) (Figure R2). Therefore, these results have to be interpreted with caution, and we have included them in the response only to not mislead readers. As you can see, the correlation in IES retention is moderate with Fire1/2 and ISWI1 single silencings (Figure R2A). When examining the overlaps in IES retention, the double-silencing does not fully encompass Fire1/2 and ISWI1, but it also does not have many additional IESs that are retained but not shared with Fire1/2 or ISWI1 (Figure R2B).

Based on the results from these double knockdowns, we believe that Fire1/2, TFIIIS4 and ISWI1 are in the same pathway. Moreover, the H3K9me3 and H3K27me3 IF data suggests that May1-4 may be downstream of Fire1/2, which can explain why most of the May1-4-dependent IESs are also Fire1/2-dependent.

Figure R2. Shared IES retention between Fire1/2, ISWI1 and Fire/ISWI1-KD.

(A) Correlation plots calculated by hexagonal binning of IES retention scores generated using After_ParTIES (Swart et al., 2017). Pearson's correlation coefficients are given above each subgraph. Red lines are for ordinary least-squares (OLS) regression, orange lines for LOWESS, and gray lines for orthogonal distance regression (ODR). From light green to dark blue, the correlation is stronger.

(B) Venn diagram depicting shared IES retention between Fire1/2, ISWI1 and Fire/ISWI1-KD, for IESs with an IRS > 0.1. **(C)** IES retention PCRs of macronuclear DNA extracted from Fire/May, Fire/TFIIIS4 and Fire/ISWI1-KD. **(D)** IES retention PCRs of macronuclear DNA extracted from three replicates of Fire/ISWI1-KD. Replicate 1 was sent for deep sequencing.

•It is very important to do ChIP-seq for TFIIIS4, ISWI1, Fire and May. The above system is well understood in flies because the genomic localization of individual players is clearly evident. If this

important experiment does not seem feasible, the conclusions that can be drawn should be much weaker.

As explained in depth above, this experiment is unfortunately not feasible. We have therefore carefully revised the manuscript to tone down the statements where appropriate.

- There are no or very weak data to support that H3K27me3 promotes transcription.

Despite our best attempts, we have not been able to determine whether Fire1/2 binds H3K27me3 nor if H3K27me3 is directly linked to TFIIIS4-dependent transcription. We have therefore toned down the statements throughout the manuscript to better fit our data by focusing on the PRC2 complex and Fire1/2 rather than overstating H3K27me3. Instead, we have discussed the preliminary evidence of H3K27me3 acting as an activating mark in the discussion section.

- Many of the figures are unreadable due to improper font size.

We have increased the font sizes.

- What are the units for the vertical axis in Fig. 1B? In the selection of subfamily, there is a threshold of <500, but no numerical evidence is given as to whether this is a low expression or not.

The vertical axis is the mean expression level of each gene at each given timepoint, generated from published DESeq2-normalized RNA-seq counts (Arnaiz et al., 2017) retrieved from the *Paramecium* Database (Arnaiz et al., 2020). As for the threshold, we chose this based on our previous experiences working with lowly expressed genes during autogamy, where we had problems knocking down genes with less than < 500 expression in the database or observing GFP signals using constructs containing these genes. We have added the axis label and included this information in the figure legend.

- The horizontal axis in Fig. 1B is unclear because it is not written in the legend.

Thank you – we have added the information to the legend.

- In Fig. 2C, the text states that it is looking at the localization of Fire2, but there is no data.

Thank you - we have performed GFP localiation of Fire2 and added the results to the manuscript.

- Fig.2D, the quality of the photo in FigureS2 is poor, and it is not clear if these condensates are condensates or not. In regard to the sentences below, "These condensates appear dynamic and display variable sizes that at first are small, then increasing in size and decreasing in number as development progresses, before only a few condensates are left after karyonidal division, which eventually disappear altogether", the corresponding figure does not support this claim.

Thank you for the comment. Firstly, regarding the "condensate" term, as both Reviewer #2 and #3 pointed out, we have not proven that these are in fact condensates and have therefore changed the term to "foci". Secondly, we agree that the quality of the pictures is not ideal for demonstrating the dynamics of these foci as they were taken directly after applying EDTA and DAPI staining in cell droplets, without any treatment or washing, and on a suboptimal microscope. Since the dynamics of

May1 foci is not the main focus of our study, and the foci are in particular difficult to see in the images of later stages of development, we have kept only the top two images in the figure and have modified the text accordingly to state "These foci appear dynamic and display variable sizes that at first are small, then increasing in size and decreasing in number as development progresses."

- The analysis in Fig. 2C, D, G, and H should be performed for all paralogs of Firefly and Mayfly. We have included images of Firefly2 to the supplement to have a complete set of ohnologs for the main focus of the paper. We do not think this is necessary for Mayfly, given the high sequence similarity between Mayfly ohnologs. For G and H, we have already tried extensively to purify Fire proteins from *Paramecium* lysate for Co-IP, ChIP, or other binding assays. These attempts unfortunately failed, as the protein was not soluble in any condition we tried. We then went on to re-codonize Fire1 for protein expression in *E. coli*, which also failed due to solubility issues, as well as *in vitro* protein expression kits (TnT® Quick Coupled Transcription/Translation System and S30 T7 High-Yield Protein Expression System from Promega), without success. We ask for the reviewers understanding in this regard.

- As for the data in Fig. 2E, I could not understand the reason why there are Dead, Sick, and Healthy phenotypes. I would also like to see a cell picture that would be a representative example for each phenotype.

This is a standard assay in the *Paramecium* field and is included in many papers (e.g., PMID: 28283070, PMID: 26177014, PMID: 36001962, PMID: 19884254). As is stated in the methods section, it is based on the post-autogamous division rate compared to wild-type cells, and the morphology of the cell doesn't change (nor can we provide a picture of a dead cell, they rupture and disappear). We have included a better explanation of this assay in the revised manuscript to clarify.

- p.6 "There was also a very faint signal in the maternal MAC at early stages of development, but much weaker than the new MAC signal and often As the proteins shine bright in the developing new MACs only for a short time window before fading, the proteins were named Firefly (Fire)." There is no data on this claim.

Because the signal is so faint and inconsistent in the maternal MAC, we believe that it is better to not include this and simply state it, since images showing this may be misleading to the readers. We have included images of the later stages of new MAC development to the supplement.

- In the analysis of small RNAs from Fig. 3 onward, the authors should add the analysis of sequence contents and targets. The length distribution data alone is not enough to support the author's claim.. We are confused about this comment because the figure is not simply the length distribution of small RNAs, the colors correspond to the targets and this is a very common way to show small RNA data in the field. We are even showing more targets than most previously published papers because we wanted to be comprehensive. However, we agree that the legend should state this better, so to make

it more understandable for all readers, we have explained it better in the legend. We have also included an additional sentence about it in the results section.

- The vertical axis of Fig. 3A is up to 5000, which is extremely unnatural, and it seems as if the data has been manipulated.

We have certainly not manipulated the data and we have also provided the raw sequencing data we used for the analysis as well as the processed IRSs as a table (Table EV2) The axis has been truncated to visualize the distribution of retained IESs, which is also stated in the legend “Histogram bars with more than 5,000 IESs have been truncated for clarity.”. However, we agree that this should be understandable from the figure itself, and have therefore shown this more clearly on the vertical axis of the figure. We have also modified the axes of Figure 4A and 5B in the same way.

- I do not understand the biological meaning of adopting a cutoff of $IRS > 0.1$ in the analysis of Fig.3A. Does the cell die if this value is exceeded?

We chose a cutoff of $IRS > 0.1$ in the analyses to exclude noise coming from weakly retained IESs, and better compare the “true” phenotypes of each knockdown. For each IES in the genome, an IRS from 0 to 1 is a measurement of 0 to 100% retention in the sequencing, so how many reads include vs. does not include the IES. Even when doing control KDs, some IESs can be retained very weakly due to stochastic reasons, though these aberrant retention events typically only occur in some of the copies of the genome (the ploidy of which is $800n$) or in very few cells. Therefore, to perform a meaningful analysis, such noise should be removed, which is why we only consider IESs with at least 10% retention. However, it is worth noting that having no cutoff shows the best overlap between the sets (i.e. TFIIIS4-KD and Fire1/2-KD IES retention overlap in Figure S7), though the true IES retention may be overestimated when using $IRS > 0$. Since IESs are scattered across the genome and not all are located in essential regions, this means that even when an IESs is 100% retained ($IRS = 1$), this value does not translate to the lethality – the retention of some IESs are more detrimental than others (e.g. IESs located in essential genes). We have modified our explanation in the manuscript to clarify this.

- What does periodicity mean in the results of Fig.3C and D? Also, is there any difference in the distribution pattern of such periodicity?

The reviewer asks about a very interesting feature of IESs. There is a ~ 10 bp periodicity of IES sizes, that has previously been noted to correspond to the helical repeat of double-stranded DNA and fit a 10.2 bp sine wave (PMID: 23071448). It has been suggested to be a constraint of the excision complex which may require interactions with the same face of the double helix at both sides of the IES, which would leave it unable to excise sizes between these. Since this is an inherent feature of the IES sizes found in the genome, the distribution pattern does not change.

- Figs. 5A and B are not cited.

These citations are found on Page 12.

•Fig. 6 shows a model for a series of cascades, but there is no data at all on such genetic epistasis. This model is based not only on the data presented in the manuscript, but also on previous knowledge regarding the expression and function of the players in question. Please allow us to explain a few of the key evidence supporting this model.

First of all, their expression and localization pattern. Fire1/2 and TFIS4 (PMID: 26177014) are only present in the new MACs in a very early stage of new MAC development, which precedes that of the ISWI1 peak (PMID: 36221862). Thus, Fire1/2 and TFIS4 are likely upstream of ISWI1. Then there is evidence from nucleosome profiling. ISWI1 is a nucleosome remodeler that has been shown to be required for nucleosome depletion of IESs. Thus, its depletion causes increased nucleosome occupancy over IESs. Anything upstream of ISWI1 but in the same pathway is therefore expected to also be required for nucleosome depletion, whereas anything downstream should not. Silencing of Fire1/2, the PRC2 complex (PMID: 36001962), or interfering with the scnRNA pathway (PMID: 36221862), also leads to higher nucleosome densities comparable with that of ISWI1. This places these three in the same pathway, upstream of the activities of ISWI1. Sequence-recognition of IESs occurs through scnRNAs binding to nascent transcripts much like in other nuclear sRNA systems, which places scnRNAs downstream of TFIS4, since TFIS4 is required for ncRNA transcription (PMID: 26177014). The excisase Pgm does not lead to increased nucleosome occupancies across IESs, placing Pgm downstream of ISWI1. We also know that Pgm cannot remove these IESs when any of the upstream players are silenced (PRC2, Fire1/2, TFIS4, ISWI1, scnRNAs), placing Pgm at the end.

Referee #3:

In this work, Solberg et al identify new members of chromodomain-containing protein family in *Paramecium* and conduct the detailed analyses of their role in DNA elimination. Authors find that May and Fire proteins appear to have overlapping but distinct targets. The manuscript is well written, but the main conclusion is just one of the possible explanations of the data. Most importantly, direct evidence for specific recognition of H3K27me3 by Fire1/2 should be presented. In silico prediction is informative yet insufficient to support the model in Fig.6. Genetic tests are apparently unfeasible, as Ezl1 deposits methyl groups on both K9 and K27. Instead, the authors could do in vitro experiments with recombinant proteins, as they did for May1 protein. If that is impossible, authors must present alternative models. A little suggestion: according to the Cambridge Dictionary, "condensate" means "liquid formed by condensation"; so, until liquid properties of the May or Fire protein aggregates are demonstrated, another term should be used, e.g., puncta, or anything else authors deem suitable.

As the reviewer correctly noted, genetic tests using Ezl1 are not feasible due to its dual methyltransferase activity, and unfortunately, in vitro experiments using recombinant Fire1 also did not work. We tried extensively to purify Fire proteins from *Paramecium* lysate for binding assays, which failed because the protein was not soluble in any condition we tried. We re-codonized Fire1 for protein expression in *E. coli*, but this also failed due to solubility issues, as well as *in vitro* protein expression kits (TnT® Quick Coupled Transcription/Translation System and S30 T7 High-Yield Protein Expression System from Promega), without success. We have therefore presented alternative models, as the reviewer suggested. One of these possible models is if Fire proteins have dual binding preferences, and can recognize both H3K9me3 and H3K27me3, which would make part of its activities redundant to May proteins. From the immunofluorescence data in EV control and Fire1/2 silencing, the signal of both modifications disappeared from the new MACs specifically, which could either mean that Fire1/2 binds both marks, or that Fire1/2 binds H3K27me3 which is then needed for the deposition of H3K9me3. Although the retention of many IESs are shared between Fire and May silencings, if Fire and May proteins both recognized H3K9me3 and were partly redundant, we would likely see an increased number of IESs affected by their double knockdown ("Fire KD and May KD"), which we did not observe. We therefore favor the model that Fire1/2 binds H3K27me3 which is then needed for the deposition of H3K9me3, which is read by May proteins. We have included these results and discussions to the manuscript.

In the model in Figure 6, we have now removed H3K27me3, since we have not directly proven this, and instead describe this as "The PRC2 complex sets H3K9me3 and H3K27me3 in the developing new MACs, one or both of which is read by the Pc-family proteins Fire1/2."

As for the term "condensates" we agree that we cannot call them condensates without further evidence and have changed the term to "foci".

Dear Marius,

Thank you for the transfer of your revised manuscript to EMBO reports. We have now received the enclosed reports from the referees that were asked to assess it, and I am happy to say that both support its publication now.

Only a few editorial requests will need to be addressed before we can proceed with the official acceptance of your manuscript.

- Please move the Data Availability Section to before the Acknowledgments
- Please remove the author credits from the ms file. All credits need to be entered during online ms submission.
- Please remove "DATA NOT SHOWN" that appears twice on p17, as per journal policy.
- In the author checklist, all questions on statistics need to be answered.
- Some funding info is missing in our online submission system, please add: ERC grants 260358 "EPIGENOME" and 681178 "G-EDIT"; Swiss National Science Foundation Grants 31003A_146257 and 31003A_166407; the National Center of Competence in Research (NCCR) RNA and Disease; the World Premier International Research Center Initiative (WPI), MEXT, Japan.
- Please add a currently missing callout for Figure 2E.
- The legends for the EV tables need to be removed from the ms and provided in each table file; Table EV1 is a dataset and needs to be called Dataset EV1. The following needs to be updated: source file name, title, ms callout; the legend should be provided as a separate sheet/tab in the Excel file.
- As far as I understand, Tables EV2 and 3 are also Datasets. Please also call them Dataset EV2 and Dataset EV3 and add the legends to the table files and update the ms callouts.
- The APPENDIX FILE needs to be cleaned (without highlighting, track changes, etc.); page numbers are needed for the table of content on the title page.
- The synopsis image is OK but I think the quality would be better if you upload an image file, such as jpg or tiff. Please upload it at the final size of 550x300 pixels.
- Please upload the Research and Tools table as a separate file.
- Figure 2D & Appendix Fig S4 seem to show the same image, but this is not mentioned in the figure legends. Please clarify and correct.
- Please note that the exact p values are not provided in the legend of figure 5E.
- Please note that for the figures 5E, p-values and statistical tests are indicated in the legends. However, comparison for the same, ""*****"" has not been represented in the figures. Please rectify this in the figures as applicable.
- Please note that the box plots need to be defined in terms of minima, maxima, centre, bounds of box and whiskers, and percentile in the legend of figure 5E
- Please note that information related to n is missing in the legends of figures 5E, EV4 B.
- Please note that for heatmap present in figures 3D, 4C, EV3 A, a numbered scale bar is not provided. This needs to be rectified."
- Please note that the scale bar is missing for figure 4D.
- Please note that scale bar and its definition are missing for figures 2C, D.

I would like to suggest some minor changes to the abstract. Please check if all is correct and let me know whether you agree with the following:

Facultative heterochromatin is marked by the repressive histone modification H3K27me3 in eukaryotes. Deposited by the PRC2

complex, H3K27me3 is essential for regulating gene expression during development and chromatin bearing this mark is generally considered transcriptionally inert. The PRC2 complex has also been linked to programmed DNA elimination during development in ciliates such as Paramecium. Due to a lack of mechanistic insight, a direct involvement has been questioned as most eliminated DNA segments in Paramecium are shorter than the size of a nucleosome. Here, we identify two sets of histone methylation readers essential for PRC2-mediated DNA elimination in Paramecium: Firefly1/2 and Mayfly1-4. The chromodomain proteins Firefly1/2 act in tight association with TFIIS4, a transcription elongation factor required for non-coding RNA transcription. These noncoding transcripts [OK?] act as scaffolds for sequence-specific targeting by PIWI-bound sRNAs, resulting in local nucleosome depletion and DNA elimination. Our findings elucidate the molecular mechanism and the role of PRC2 in PIWI-mediated DNA elimination and suggest that its role in IES elimination may be to activate rather than repress transcription.

Referee #1:

This paper is a first submission to EMBO Reports of a version which was apparently transferred from EBMO J to EMBO Reports. Since I have already reviewed the paper there and the authors respond to this review, I am now treating this submission as a revision.

The authors have significantly improved the manuscript and addressed my concerns. The final model has also been improved by alternatives in response to referee 3's comment, and the text in the conclusions and throughout the manuscript has been improved, and some statements that were driven but too far have been smoothed out. I appreciate the additional analyses carried out by the authors regarding the contribution of Fire1/2 dependent and independent IESs and IF quantification of H3K27me3. The results are included and discussed in the manuscript. I am happy with the current version being convinced the first description of these two new candidates results in new insight into this exciting mechanism of programmed DNA elimination.

Referee #2:

The authors have addressed the main concerns that arose in the previous round of review. The manuscript contains new data and its interpretations are adequate. The manuscript should be accepted at EMBO Reports.

All editorial and formatting issues were resolved by the authors.

Prof. Mariusz Nowacki
University of Bern
Institute of Cell Biology
Baltzerstrasse 4
Bern, BE 3012
Switzerland

Dear Mariusz,

I am very pleased to accept your manuscript for publication in the next available issue of EMBO reports. Thank you for your contribution to our journal.
